# Nylons with Highly-Bright and Ultralong Organic Room-Temperature Phosphorescence

Dian-Xue Ma[1,2,3,4], Zhong-Qiu Li[1,2,3], Kun Tang [1,2,3], Zhong-Liang Gong [1,2,3], Jiang-Yang Shao [1,2,3] & Yu-Wu Zhong [1,2,3,4] ✉

Endowing the widely-used synthetic polymer nylon with high-performance organic room-temperature phosphorescence would produce advanced materials with a great potential for applications in daily life and industry. One key to achieving this goal is to find a suitable organic luminophore that can access the triplet excited state with the aid of the nylon matrix by controlling the matrix-luminophore interaction. Herein we report highly-efficient room-temperature phosphorescence nylons by doping cyano-substituted benzimidazole derivatives into the nylon 6 matrix. These homogeneously doped materials show ultralong phosphorescence lifetimes of up to 1.5 s and high phosphorescence quantum efficiency of up to 48.3% at the same time. The synergistic effect of the homogeneous dopant distribution via hydrogen bonding interaction, the rigid environment of the matrix polymer, and the potential energy transfer between doped luminophores and nylon is important for achieving the high-performance room-temperature phosphorescence, as supported by combined experimental and theoretical results with control compounds and various polymeric matrices. One-dimensional optical fibers are prepared from these doped room-temperature phosphorescence nylons that can transport both blue fluorescent and green afterglow photonic signals across the millimeter distance without significant optical attenuation. The potential applications of these phosphorescent materials in dual information encryption and rewritable recording are illustrated.

Organic room-temperature phosphorescence (RTP) materials with long-lived lifetimes have been intensively investigated in recent years, which hold great promise for applications in the multidisciplinary fields of material science and technology[1], information protection[2,3], biological visualization and imaging[4,5], anticounterfeiting[6,7] and various optoelectronic devices[8,9]. Though much progress has been achieved in this field, it remains a challenge to obtain highly efficient organic RTP materials with both long phosphorescence lifetimes ($\tau_P$) and high phosphorescence quantum yields ($\Phi_P$). This challenge is

mainly limited by the weak spin-orbit coupling (SOC) of common organic materials and the unavoidable dissipation of triplet excitons as a result of molecular motions and the exciton quenching by the surrounding environment[10,11]. To meet this challenge, some rational strategies have been proposed and implemented to generate and stabilize triplet excitons under ambient conditions to achieve efficient RTP. For instance, the intersystem crossing (ISC) transition can be facilitated by the use of heavy atoms and heteroatoms with lone-pair electrons[12]. The creation of a rigid environment through crystal engineering[13,14],

[1]Key Laboratory of Photochemistry, Institute of Chemistry, Chinese Academy of Sciences, Beijing, China. [2]Beijing National Laboratory for Molecular Sciences, Beijing, China. [3]CAS Research/Education Center for Excellence in Molecular Sciences, Institute of Chemistry, Chinese Academy of Sciences, Beijing, China. [4]School of Chemical Sciences, University of Chinese Academy of Sciences, Beijing, China. ✉e-mail: zhongyuwu@iccas.ac.cn

host-guest interaction[15–17], polymerization[18,19], and supramolecular assembly would decrease the triplet nonradiative decay rate ($k_{P,nr}$), leading to the improvement of $\Phi_P$ and $\tau_P$[20–22].

Among various efficient RTP materials developed to date, organic crystals sometimes suffer from poor processability and flexibility. Polymer-based long-lived RTP is appealing in terms of processability and flexibility. The long chains and intertwined structures of polymers give rise to the rigid environment with multiple intermolecular interactions, which protect triplet excitons from the phosphorescence quenching by surroundings and are beneficial for realizing ultralong and bright RTP. Therefore, a series of research advancements have been achieved in polymer-based RTP materials in recent years[23–25]. In particular, polyvinyl alcohol (PVA) and poly (methyl methacrylate) (PMMA) have become the standard matrixes to design RTP polymer materials by doping with different types of organic luminophores[26]. Nevertheless, the development of RTP materials with both long $\tau_P$ and high $\Phi_P$ under ambient conditions remains a challenging task.

Nylons (or polyamides) are important polymer materials that are widely used in industry and daily life, including textile, automotive, military equipment, and packagings[27,28]. Nylons are typically produced in the form of films or fibers, representing the first type of mass-produced thermoplastic engineering plastics. Among the many kinds of commercially available nylons, the semicrystalline nylon 6 (polyamide 6; PA6) is a significant prototype polymer in an industry that possesses a number of favorable properties, including good chemical and aging resistance[29,30], appropriate thermal and electrical resistance[31], and excellent mechanical properties[32]. These properties can be partially attributed to the presence of multiple hydrogen bonding interactions among the amide groups of the polymer backbones, which effectively restrict the molecular motions in the solid state. Considering the wide applications of nylons and the appealing feature of RTP, endowing nylons with highly efficient RTP properties would produce a class of advanced materials with great potential for practical applications. On the one hand, nylons can create a rigid network through intermolecular hydrogen bond interactions among polymeric chains. On the other hand, the rich carbonyl and amino groups of these polymers allow them to form abundant hydrogen bond interactions with doped luminophore molecules, thereby effectively inhibiting the non-radiative transition of luminophore and improving the phosphorescence performance. In spite of these features, it is surprising that only very limited examples of RTP nylons with moderate performance have been reported to date. For instance, the melt mixture of PA6 and a two-dimensional polyamide sheet exhibits RTP with $\Phi_P$ of 1.8% and $\tau_P$ of shorter than 10 ms[33]. PA6 doped with triphenylamine boric acid shows RTP with $\Phi_P$ of 14.7% and $\tau_P$ of 724 ms[34]. The RTP properties of these reported doped nylons are much inferior to those of the state-of-the-art RTP organic crystals and polymeric materials[1,21,35–37]. It is of high significance and urgency to develop an excellent type of luminophore dopant that is able to achieve highly efficient RTP with nylons.

We present herein cyano-substituted phenylbenzimidazole derivatives as the luminescent luminophore dopants and nylon 6 and related polyamides as the polymer matrix to develop RTP materials with both high $\Phi_P$ and long $\tau_P$. The fusion of imidazole with aromatic rings produces the low energy (n, π*) state as a result of the transition from the imidazole nitrogen atom to the aromatic ring π system. These molecules are potentially phosphorescent via the $^1(\pi, \pi^*) \rightarrow {}^3(n, \pi^*)$ and/or $^1(n, \pi^*) \rightarrow {}^3(\pi, \pi^*)$ exciton transitions[38]. In addition, the cyano and imidazole groups enable the formation of multiple intermolecular hydrogen-bonding interactions with the nylon matrix, further benefiting the suppression of non-radiative transitions. The efficient energy transfer from the nylon matrix to the luminophore guest contributes to the excellent RTP performance. Ultralong phosphorescences are obtained from nylon 6 films doped with these cyano-substituted phenylbenzimidazole derivatives under ambient conditions after

photoexcitation. These doped films show $\tau_P$ of up to 1.5 s and high $\Phi_P$ of up to 48.3% at the same time, exhibiting strong green afterglow (lasting for 18 s) visible to the naked eye. Furthermore, benefiting from the facile processability of nylons, one-dimensional (1D) optical fibers are prepared that can transport both fluorescent and afterglow phosphorescent photonic signals in the mm range without significant optical attenuation. The potential applications of these phosphorescent materials in information encryption and rewritable paper are further demonstrated.

## Results
### Material synthesis
The luminophore 2-phenyl-1H-4,7-dicyanobenzo[d]imidazole (**1**) was obtained from the condensation of 2,3-diaminoterephthalonitrile with benzaldehyde in 88% yield (see details in the Supplementary Information). The synthetic methods of other derivatives are similar to that of **1**, including 2-phenyl-1H-5,6-dicyanobenzo[d]imidazole (**2**), 2-methyl-1H-4,7-dicyanobenzo[d]imidazole (**3**), 2-(4′-methoxy-[1,1′-biphenyl]−4-yl)−1H-4,7-dicyanobenzo[d]imidazole (**4**), and 2-phenyl-1-methyl-4,7-dicyanobenzo[d]imidazole (**6**) (see details in the Supplementary Information). Compounds 2-phenyl-1H-benzo[d]imidazole (**5**) and 1,4-dicyanobenzene (**7**) and nylon samples are commercially available. The structures of these cyano-substituted phenylbenzimidazole derivatives were fully characterized by nuclear magnetic resonance spectra ($^1$H and $^{13}$C NMR), high-resolution mass spectrometry (HRMS), and element analysis (Supplementary Figs. 1 to 16). The doped nylon samples were obtained by grinding a mixture of nylon and luminophore, followed by melting at a suitable temperature (see details in the Supplementary Information) and subsequent fast (in 10 s) or slow cooling (annealing in 30 min) to rt (Fig. 1).

### Nylon 6 doped with luminophore 1
Firstly, we selected compound **1** as a model emitter and nylon 6 as the matrix to investigate the photophysical properties of the doped film named **1@PA6** (Fig. 1). The melt film of **1@PA6** prepared by a fast cooling method with a 0.1 wt% doping ratio is blue-emissive under the irradiation of a UV lamp. It shows green afterglow when the lamp is turned off, which lasts for 18 s as detected by the naked eye (Fig. 1f and Supplementary Movie 1). As shown in Fig. 1b, the emission maximum of the steady-state photoluminescence (PL) spectrum of **1@PA6** locates at 420 nm ($\lambda_F$) and it has a short lifetime of 4.0 ns ($\tau_F$), suggestive of a typical fluorescence behavior (Supplementary Fig. 17). After a delay time of 5 ms, a structureless emission band appears with the emission maximum at 510 nm ($\lambda_P$) and an impressively long lifetime of up to 1.51 s ($\tau_P$) under ambient conditions, demonstrating a long-persistent RTP character (Fig. 1c). As shown in Supplementary Fig. 18, the shapes of the fluorescence and phosphorescence spectra of **1@PA6** are essentially independent on the measurement condition (in air, under vacuum, or after being exposed to $O_2$). The doped polymer does not display white emission under UV excitation, probably due to the significant difference in the luminescence and phosphorescence lifetime. In addition, the phosphorescence lifetime of the **1@PA6** film is almost the same under different measurement conditions, suggesting the excellent oxygen-shielding effect of the polymer matrix.

The absolute $\Phi_P$ of **1@PA6** reaches 48.3%, while its fluorescence quantum efficiency ($\Phi_F$) is 47.8%. When the excitation wavelength is varied from 280 to 380 nm, the shape of the phosphorescence emission band essentially remains unchanged without distinct hypsochromic or bathochromic shift, indicating that the phosphorescence emission is independent on the excitation wavelength (Supplementary Fig. 19). The profile of the phosphorescence spectrum of **1@PA6** is essentially consistent with that of **1** in dilute solution at 77 K (Supplementary Fig. 20), indicating that the emitter **1** is homogeneously distributed within the nylon 6 matrix. In addition, the phosphorescence properties are sensitive to temperature (Supplementary

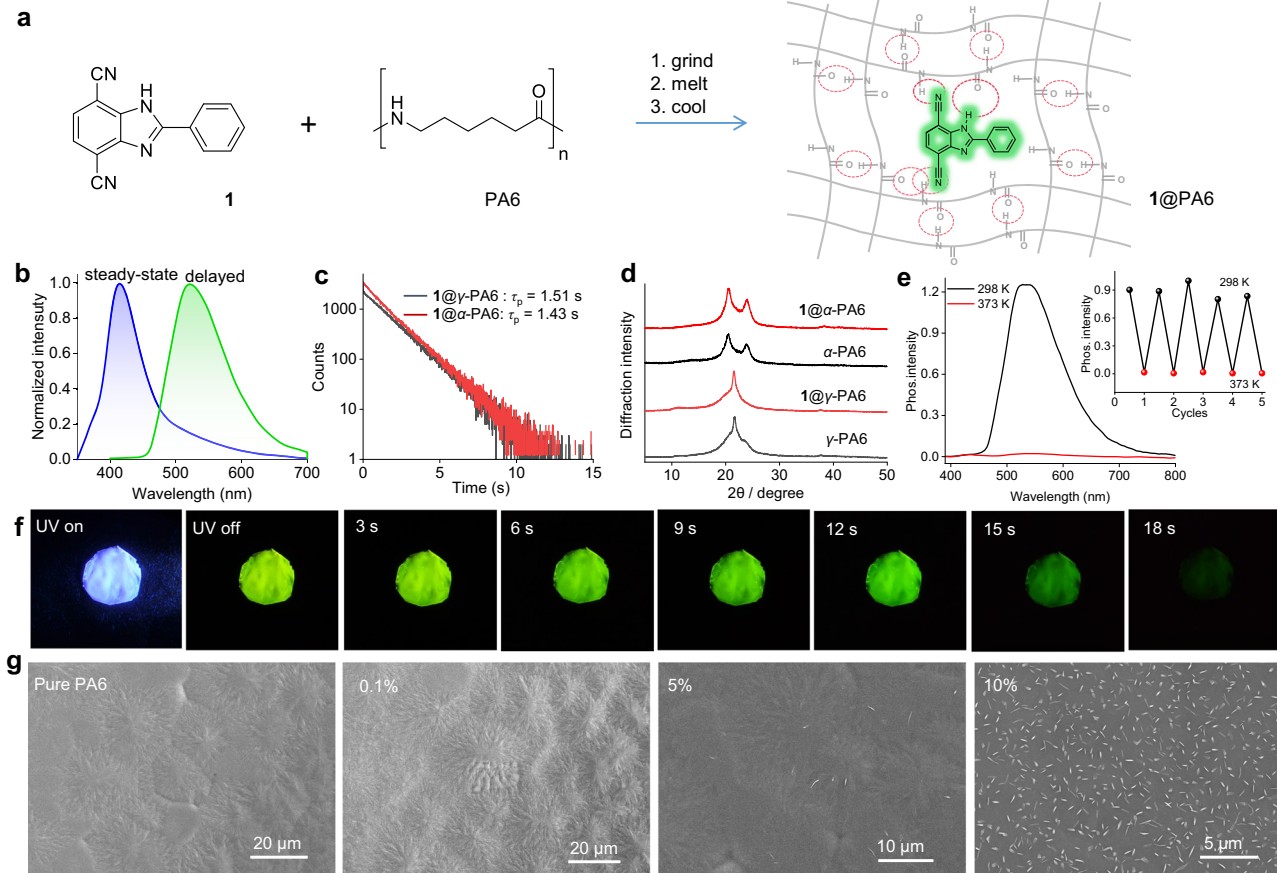

**Fig. 1 | Studies on nylon 6 doped with compound 1. a** Schematic illustration of the preparation of **1@PA6**. **b** Normalized steady-state (blue line) and delayed emission spectra (green line) in air of 0.1% **1@γ-PA6** film excited at 320 nm. Delay time: 5 ms. **c** Lifetime decay curve for the phosphorescence at 510 nm of γ and α phase PA6 doped with 0.1% of **1**. **d** PXRD spectra of γ and α phase PA6 with or without 0.1% of **1**. **e** Phosphorescence spectra of 0.1% **1@γ-PA6** at 298 and 373 K. Inset: the changes of phosphorescence intensity upon five heating-cooling cycles. **f** Photographs of the long-lived phosphorescence emission of 0.1% **1@γ-PA6** before and after turning off the UV excitation from 0 to 18 s. **g** SEM images of pure γ-PA6 and **1@γ-PA6** film with doping concentration of 0.1, 5, and 10 wt%, respectively.

Figs. 21 and 22). The phosphorescence intensity and lifetime of **1@PA6** show a decreasing trend upon increasing the temperature from 77 to 377 K, ruling out the assignment of the green emission to thermally activated delayed fluorescence. The phosphorescence lifetime of the doped film decreases sharply to 7.16 ms at 373 K. When the temperature is further decreased to room temperature, the phosphorescence performance of the film is restored to its initial state, and this process can be repeated by multiple times, suggesting the high thermal and photostability of this doped material (Fig. 1e).

These nylon films were all obtained by a fast cooling method. The powder X-ray diffraction (PXRD) analysis shows that these samples, either with or without the dopant of **1**, belong to γ-phase nylons, as supported by the appearance of a distinct diffraction peak at 2θ of 21.5° (Fig. 1d). This peak is indexed to the overlapping (200) and (001) reflections of γ-phase nylons[39]. We have also prepared the α-phase nylons in the absence or presence of **1** by slowly cooling the melted sample. The α-phase nylons display two peaks at 2θ of 20.5° and 23.9°, attributed to the (200) and (002)/(202) diffraction, respectively[40]. The doping of a small amount of **1** essentially does not change the crystallinity of nylons. In addition, γ-phase and α-phase nylons doping with 0.1% of **1** possess similar RTP properties. The film of **1@a-PA6** has a $\tau_P$ of 1.43 s, comparable to that of **1@γ-PA6** (Fig. 1c). Considering that γ-phase nylons can be prepared in a shorter time, they are used as the matrix for other studies in this work, unless otherwise noted.

We further investigated the influence of the doping ratio of **1** versus the polymer matrix on the phosphorescence performance at rt. With the doping concentration of **1** increasing from 0.1 to 10 wt%, both $\Phi_p$ and $\tau_P$ decrease distinctly (Supplementary Fig. 23 and Supplementary Table 1). When a smaller doping ratio, e.g., 0.05% or 0.01%, is adopted, the RTP performance cannot be further improved. The scanning electron microscopy (SEM) analysis shows that the surface of the pure PA6 sample is characterized with some spherulitic microstructures with a size of a few tens of μm (Fig. 1g). This particular morphology rich in elongated crystalline lamellae growing from a nucleation center is commonly observed for crystalline polyamides[41]. When PA6 is doped with 0.1% of **1**, the micro-spherulitic morphology does not show a distinct change, suggestive of a homogenous distribution of **1** in the polymer matrix. In contrast, when the doping ratio is increased to 5% or 10% for the sample with poor RTP performance, some needle-like aggregates appear on the film surface. These results suggest that the homogeneous distribution of luminophore dopant in the polymer matrix is critical for maintaining highly-efficient RTP, as a result of the sufficient matrix-guest interaction.

## Expansion of luminophore substrates and nylons

To validate the generality of our strategy, we designed and synthesized three other luminophores, **2, 3,** and **4**, containing the 1H-benzo[d]imidazole core with two cyano substituents (Fig. 2a). Compound **2** has two cyano substituents on the 5,6-positions of the 1H-benzo[d]imidazole core, in comparison to compound **1** containing two cyano substituents on the 4,7-positions of the core. Other than a phenyl group in compound **1**, compound **3** and **4** possess a methyl and *p*-methoxy-1,1'-biphenyl group, respectively, on the 2-position of the 1H-benzo[d]imidazole core. The doping concentrations of all of these

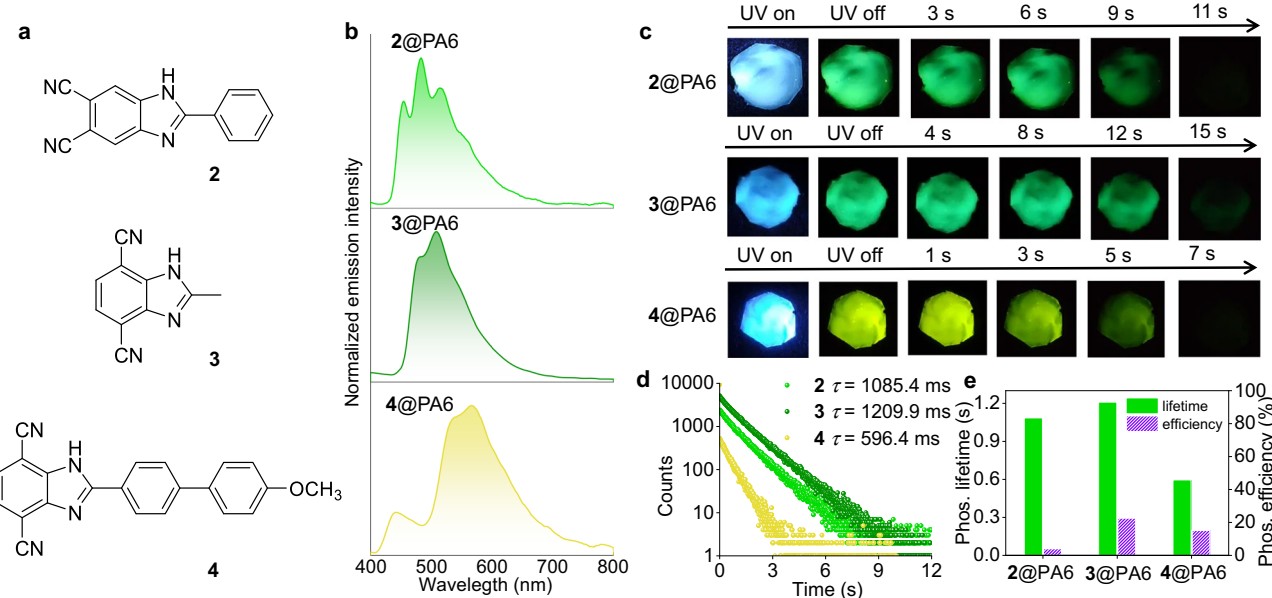

**Fig. 2 | Photophysical characterizations PA6 films doped with 0.1 wt% of 2, 3, and 4 under ambient conditions. a** Chemical structures of luminophores **2**, **3** and **4**. **b** Normalized phosphorescence spectra of **2**@PA6, **3**@PA6 and **4**@PA6. **c** Photographs of the long-lived luminescence **2**@PA6, **3**@PA6, and **4**@PA6 films taken before and after turning off UV excitation. Excitation power: 10 W. Exposure time: 10 s. **d** Lifetime decay profiles of phosphorescence emission at 490 nm for **2**@PA6, 500 nm for **3**@PA6, and 550 nm for **4**@PA6. **e** A comparison of measured phosphorescence (Phos.) lifetime and efficiency.

## Table 1 | Key photophysical data of doped polymers

| Material[a] | $\lambda_F$ (nm)[b] | $\tau_F$ (ns)[c] | $\Phi_F$ (%)[d] | $\lambda_P$ (nm)[e] | $\tau_P$ (ms)[f] | $\Phi_P$ (%)[g] | $k_{P,r}$ (s⁻¹)[h] | $k_{P,nr}$ (s⁻¹)[i] | $\Phi_{isc}$(%)[j] | $k_{isc}$ (s⁻¹)[k] |
|---|---|---|---|---|---|---|---|---|---|---|
| 1@PA6 | 420 | 4.0 | 47.8 | 510 | 1515.5 | 48.3 | 0.32 | 0.34 | 50.3 | $6.0 \times 10^7$ |
| 2@PA6 | 360 | 1.5 | 35.6 | 490 | 1085.4 | 5.3 | 0.05 | 0.95 | 13.0 | $3.1 \times 10^7$ |
| 3@PA6 | 390 | 4.2 | 35.3 | 500 | 1209.9 | 29.5 | 0.24 | 0.58 | 45.5 | $3.8 \times 10^7$ |
| 4@PA6 | 440 | 2.4 | 79.1 | 550 | 596.4 | 19.8 | 0.33 | 1.34 | 20.0 | $6.6 \times 10^7$ |
| 1@PA6/6 | 410 | 3.2 | 10.7 | 510 | 1254.9 | 6.4 | 0.05 | 0.95 | 37.4 | $1.3 \times 10^7$ |
| 1@PA6/10 | 400 | 3.2 | 25.2 | 510 | 1173.3 | 10.3 | 0.09 | 0.76 | 29.0 | $2.3 \times 10^7$ |
| 1@PA12 | 410 | 3.0 | 35.7 | 510 | 5.6 | <0.1 | -- | -- | -- | -- |
| 5@PA6 | 340 | 1.5 | 29.7 | 450 | 42.2 | 1.0 | 2.44 | 23.56 | 3.4 | $6.7 \times 10^6$ |
| 6@PA6 | 390 | 2.5 | 48.8 | 520 | 823.7 | 18.4 | 0.22 | 0.99 | 27.4 | $5.3 \times 10^7$ |
| 7@PA6 | 380 | 4.1 | 2.4 | 490 | 623.3 | 0.8 | 0.01 | 1.59 | 25.5 | $1.5 \times 10^6$ |
| 1@PVA | 405 | 3.0 | 39.7 | 515 | 696.1 | 0.3 | 0.004 | 1.43 | 0.73 | $9.7 \times 10^5$ |
| 1@PMMA | 385 | 2.8 | 40.2 | 490 | 3.1 | <0.1 | -- | -- | -- | -- |
| 1@PS | 405 | 2.9 | 42.5 | 490 | 4.4 | <0.1 | -- | -- | -- | -- |

[a]0.1 wt% doping ratio for all.
[b]Fluorescence emission maximum.
[c]Fluorescence decay lifetime at $\lambda_F$.
[d]Absolute fluorescence quantum yield.
[e]Phosphorescence emission maximum.
[f]Phosphorescence decay lifetime at $\lambda_P$.
[g]Absolute phosphorescence quantum yield.
[h]Phosphorescence radiative rate constant $k_{P,r} = \Phi_P/\tau_P$.
[i]Phosphorescence non-radiative rate constant $k_{P,nr} = (1-\Phi_P)/\tau_P$.
[j]ISC efficiency: $\Phi_{isc} = \Phi_P/(\Phi_P + \Phi_F)$.
[k]Rate constant of ISC: $k_{isc} = \Phi_{isc}\Phi_F/\tau_F$.

compounds are kept at 0.1 wt% with respect to the nylon 6 matrix. The doped samples were obtained by the similar melting and fast cooling procedure for the preparation of **1**@γ-PA6. The resulting **2**@PA6, **3**@PA6, **4**@PA6 films show long-lived luminescence with a duration time of 7 to 15 s, which could be clearly observed by the naked eye under ambient conditions after turning off the irradiation (Fig. 2c). The films of **2**@PA6, **3**@PA6, and **4**@PA6 possess $\lambda_F$ at 360, 390, and 440 nm, with corresponding $\tau_F$ of 1.5, 4.2, and 2.4 ns, respectively (Supplementary Fig. 24 and Table 1). In addition, they display $\lambda_P$ at 490,

500, and 550 nm (Fig. 2b), with long $\tau_P$ of 1.09, 1.21, and 0.60 s, respectively. Except for **2**@PA6 with a relatively smaller $\Phi_P$ of 5.3%, **3**@PA6 and **4**@PA6 films are characterized with a high $\Phi_P$ of 19.5% and 19.8%, respectively. The phosphorescence spectra of these films are in agreement with those of dilute solutions of **2** – **4** at 77 K (Supplementary Fig. 25). Nylons doped with **1** – **3** show green RTP, while **4**@PA6 exhibits yellow RTP due to the introduction of the substituent with a larger degree of conjugation. The films of **1**@PA6, **3**@PA6, and **4**@PA6 have a comparable phosphorescent radiative decay rate ($k_{P,r}$)

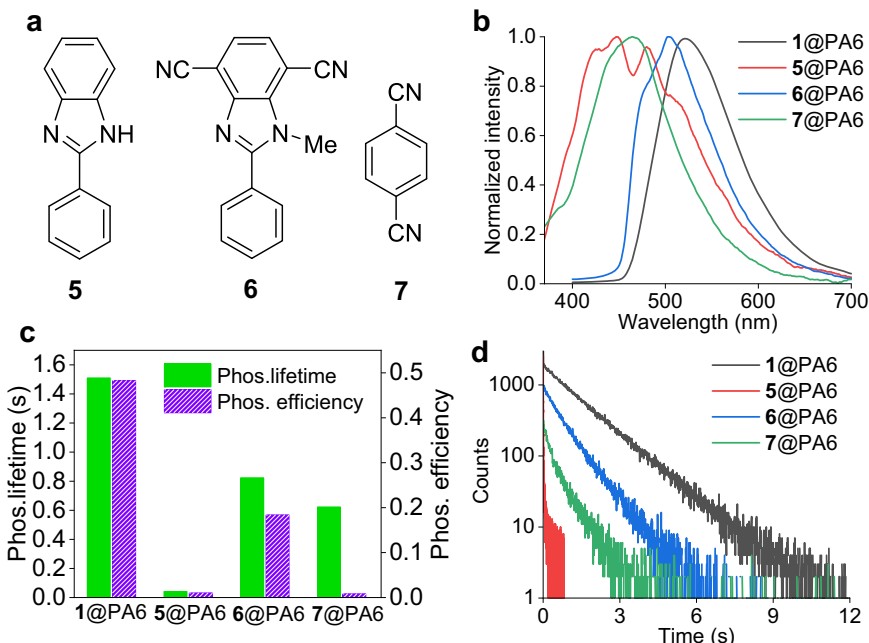

**Fig. 3 | A comparison study of PA6 films doped with 0.1 wt% of 1 and control compounds 5, 6, and 7. a** Chemical structures of control compounds. **b** Delayed emission spectra with a delay time of 5 ms. **c** A comparison of measured phosphorescence (phos.) lifetime and efficiency. **d** Lifetime decay profiles at maximum phosphorescence wavelength.

(0.24 – 0.33 s$^{-1}$, Table 1). In contrast, **2**@PA6 has a smaller $k_{P,r}$ of 0.05 s$^{-1}$. Among four composites, **1**@PA6 has the smallest $k_{P,nr}$ of 0.34 s$^{-1}$ and the highest ISC efficiency $\Phi_{isc}$ of 50.3% as estimated by $\Phi_{isc} = \Phi_P/(\Phi_P + \Phi_F)$[13]. The ISC rate constants $k_{isc}$ of these materials are in the 10$^7$ s$^{-1}$ order as estimated by $k_{isc} = \Phi_{isc}\Phi_F/\tau_F$[13].

Furthermore, other nylons other than PA6, including PA6/6, PA6/10, and PA12, were examined as the polymer matrix for luminophore **1** (0.1 wt%). The chemical structures of these nylon polymers are depicted in Supplementary Fig. 26. The obtained **1**@PA6/6 and **1**@PA6/10 films display long $\tau_P$ of over 1.0 s with $\Phi_P$ of 6.4% and 10.3%, respectively, suggesting that PA6/6 and 6/10 are also suitable polymer matrices for compound **1** to afford highly-efficient RTP materials (Supplementary Figs. 27 and 28 and Table 1). In contrast, PA12 is proved to be a poor matrix for **1**. The film of **1**@PA12 is characterized with less efficient RTP with $\tau_P$ of 5.6 ms and very low phosphorescence efficiency ($\Phi_P < 0.1\%$).

**Studies on control samples**

In order to gain further insight into the RTP effect of these doped nylon films, a series of control experiments have been conducted. Firstly, compounds **5**–**7** with a related structure as luminophore **1** were used as the dopant (Fig. 3a; 0.1% doping ratio for all). Compound **5** is lack of any cyano substituent. Compound **6** has a methyl substituent on the imidazole N-H position. Compound **7** (1,4-dicyanobenzene) has no imidazole unit. Compounds **5**–**7** all exhibit phosphorescence with $\tau_P$ of longer than 1.0 s in THF at 77 K (Supplementary Fig. 29), indicating that they are efficient phosphorescent materials at low temperatures.

The doped films of **5**@PA6 and **7**@PA6 show broad and distinctly blue-shifted phosphorescence spectra between 350 and 650 nm, with $\tau_P$ of 42.2 ms and $\Phi_P$ of 1.03% for **5**@PA6 and $\tau_P$ of 623.3 ms and $\Phi_P$ of 0.82% for **7**@PA6, respectively, indicating their relatively poor RTP performance (Fig. 3c and 3d). In contrast, the **6**@PA6 film exhibits excellent RTP with $\tau_P$ of 823.3 ms and $\Phi_P$ of 18.4%, respectively. The shape of the phosphorescence spectrum of **6**@PA6 resembles that of **1**@PA6, yet the phosphorescence maximum ($\lambda_P$) exhibits a slight blue shift to 500 nm.

Furthermore, the phosphorescence properties of **1** doped in polymers other than nylons, including PVA, PMMA, and polystyrene (PS), have been tested (at 0.1% doping ratio). The phosphorescence lifetimes and quantum yields of the obtained materials differ significantly (Supplementary Figs. 30 and 31 and Table 1). The film of **1**@PVA has $\tau_P$ of 696.1 ms, suggesting that PVA is also a potential RTP matrix of **1**. However, the $\Phi_P$ of **1**@PVA is very low (0.29%). The phosphorescence emission is relatively feeble when **1** is doped into PMMA and PS, showing a $\tau_P$ of 4.1 and 7.76 ms, respectively. In addition, their phosphorescence quantum efficiencies are too low to quantify.

**Discussion on RTP Mechanism**

The high RTP performance of **6**@PA6 and poor performance of **5**@PA6 suggest that the cyano groups of **1** are important in maintaining the high RTP performance of **1**@PA6, while the imidazole N-H unit is not indispensable. Fourier transform infrared (FTIR) spectroscopy was subsequently performed to examine the hydrogen-bonding interactions in **1**@PA6 film (Fig. 4a). In particular, the stretching vibration signal of the cyano group shifts from 2237 cm$^{-1}$ for the pure **1** solid to 2229 cm$^{-1}$ for **1**@PA6 film. The change is ascribed to the formation of a relatively rigid hydrogen-bonding network between the amide protons of nylon 6 and the CN groups of **1**. The contribution of the N-H bond of **1** to the hydrogen bonding may be insignificant, considering that the composite of **6**@PA6 with a methyl substituent also shows high RTP performance. However, the presence of hydrogen bonding alone cannot account for the excellent RTP performance of **1**@PA6. FTIR results suggest that hydrogen bonding is also present in PA6 films doped with **2** – **4** (Supplementary Fig. 32). In addition, the single crystal X-ray analysis of **1** and **2** demonstrates the involvement of the CN groups in the formation of hydrogen bonding (Supplementary Fig. 33 and Supplementary Table 2). However, the crystalline solids of **1** – **4** show very weak RTP with $\tau_P$ of shorter than 30 ms (Supplementary Figs. 34 – 36). Similarly, hydrogen bonding is also present in PVA and PA12 doped with compound **1** (Supplementary Fig. 37), and these two composites show somewhat poor RTP performance, again suggesting

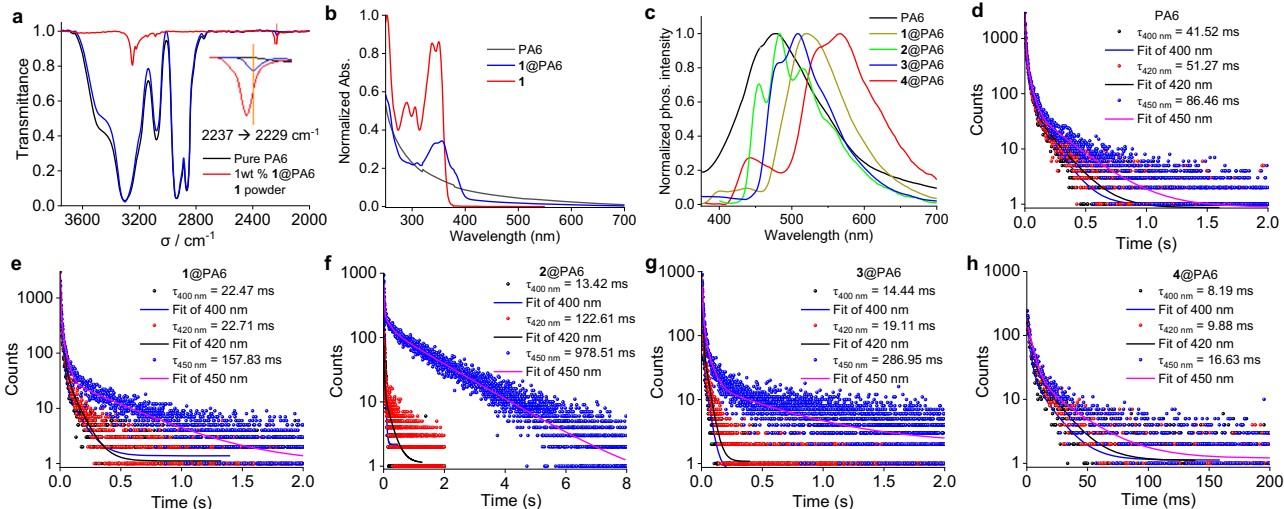

**Fig. 4 | A comparison study of PA6, 1, and PA6 films doped with 0.1 wt% of 1 – 4.** **a** FTIR spectra of PA6, **1**@PA6, and **1**. **b** Absorption spectra of PA6 and **1**@PA6 films and the solution of **1** in THF. **c** Delayed emission spectra of pure PA6 and doped PA6 films. **d,e,f,g,h** Phosphorescence lifetime decays and fitted curves at indicated wavelength of (**d**) PA6, (**e**) **1**@PA6, (**f**) **2**@PA6, (**g**) **3**@PA6, and (**h**) **4**@PA6 (excited at 290 nm).

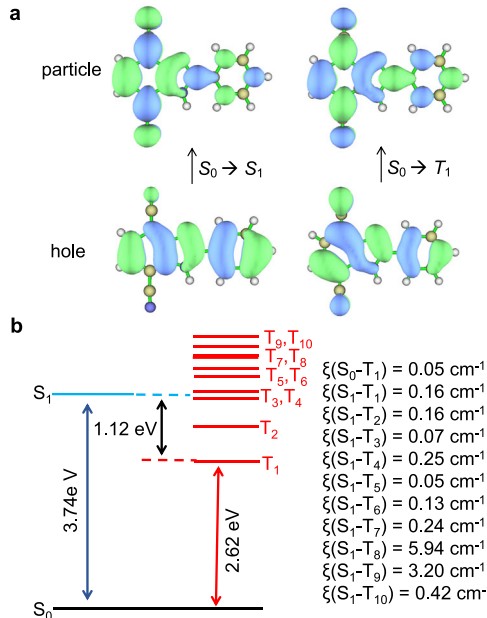

**Fig. 5 | Calculation results of 1. a** Natural transition orbitals for $S_1$ and $T_1$ states. **b** Calculated vertical excitation energies and SOC parameters.

that other factors beyond the hydrogen bonding are involved in boosting the RTP performance of **1**@PA6.

Considering that the rich amide groups of nylon polymers may assist the occurrence of clustering-triggered emission and phosphorescence[13,22], we examined the luminescence properties of the host materials. Interestingly, the pure PA6 film exhibits weak fluorescence and phosphorescence dual luminescence with $\lambda_F = 350$ nm, $\Phi_F = 0.4\%$, $\tau_F = 1.30$ ns at 350 nm and $\lambda_P = 480$ nm, $\Phi_P = 0.5\%$, $\tau_p = 119.22$ ms at 480 nm at rt (Fig. 4c, Supplementary Figs. 38 and 39). The $\tau_F$ remains essentially unchanged after doping with compound **1** (Supplementary Fig. 39). The $\tau_p$ of PA6 is wavelength-dependent, and it exhibits distinct changes after doping with **1**. In the presence of **1**, the $\tau_p$ at 400 nm is shortened from 41.52 to 21.47 ms, and that at 420 nm is shortened from 51.27 to 22.17 ms, respectively (Fig. 4d and e). This suggests that triplet-to-triplet Dexter-type energy transfer occurs from

the PA6 host to **1**, which is also supported by the presence of UV absorption of PA6 and the energy level match between PA6 and **1** (Fig. 4b and c). The $\tau_p$ at 450 nm of PA6 becomes longer after doping with **1**, which should be caused by the influence of the long-lived phosphorescence of **1**. The situation of **3**@PA6 is similar to **1**@PA6 (Fig. 4g). In the case of **2**@PA6, the $\tau_p$ at 400 nm is shorter with respect to that of PA6; however, the $\tau_p$ at 420 and 450 nm are longer than those of pure PA6 due the large overlap between the phosphorescence of **2**@PA6 and PA6 (Fig. 4f). When PA6 is doped with **4** with a red-shifted phosphorescence band, the $\tau_p$ at 400, 420, and 450 nm all become distinctly shorter (Fig. 4h). In addition to PA6, weak RTPs are observed for pure PA6/6 and PA6/10 and similar changes of $\tau_p$ as that in **1**@PA6 are observed when these polymers are doped with **1** (Supplementary Fig. 40). These data are all in support of the host-guest energy transfer of these doped polymers.

The poor RTP performance of the PA6 film containing aggregated samples of **1** (Fig. 1g and Supplementary Fig. 23) could be partially attributed to the insufficient host-guest contacts and, thus less efficient energy transfer between them. In contrast, essentially, no distinct RTP could be observed from the flexible polymer PA12, PVA, and PMMA. Though weak fluorescence and RTP are observed for PS, no potential energy transfer is evidenced in **1**@PS (Supplementary Fig. 41). The lack of efficient energy transfer in these flexible polymer composites is believed to be accountable for their poor RTP performance.

It is known that nylons are semicrystalline polymers. At this stage, we believe that the crystalline lattice size of nylon has a minor impact on the RTP performance of the current system, considering that γ- and α-phase nylons are both excellent matrix for **1**, though they have different lattice sizes[39–42]. The inferior RTP performance of **1**@PA12 is largely attributed to the lack of host-guest energy transfer.

## Theoretical calculation and discussion on photophysical properties

Theoretical calculations have been performed to help to understand the excellent RTP properties of **1**@PA6. Figure 5a shows the natural transition orbitals (NTOs) of **1** at the lowest singlet ($S_1$) and triplet ($T_1$) states. For the $S_1$ state, the hole is distributed among the phenyl-substituted benzimidazole backbone, and the particle is distributed across the entire molecule including the two cyano groups. This partial overlap of hole and particle means that the $S_1$ state has a hybrid local

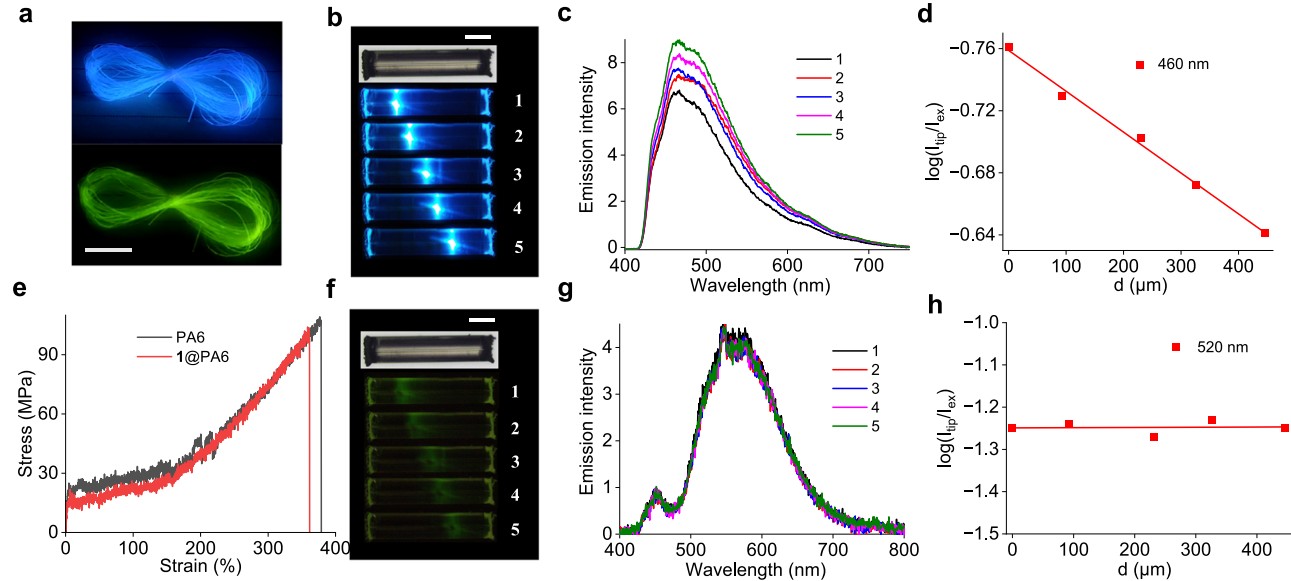

**Fig. 6 | Studies of 1@PA6 fiber. a** Image of a macroscopic bundle of fibers of 1@PA6 showing blue fluorescence and green phosphorescence, respectively. Excited at 365 nm. Scale bar: 2 cm. **b, f** Bright field and fluorescence (**b**) and phosphorescence (**f**) images of a single fiber of 1@PA6 by exciting at five different positions. Scale bar: 200 μm. **c, g** Corresponding spatially resolved fluorescence c and phosphorescence (**g**) spectra collected from the tip of the fiber. **d, h** The relation of log ($I_{tip}/I_{ex}$) vs d at the fluorescence (**d**) and phosphorescence (**h**) wavelength, where $I_{tip}$ and $I_{ex}$ are the emission intensity at the emitting tip and the excited site, respectively, and d is the light propagating distance between the excitation spot and the fiber tip. e Stress-strain test curves of PA6 and 1@PA6 fibers.

and charge transfer (HLCT) character, which is believed beneficial for the singlet to triplet ISC process. This type of luminescence is known to be sensitive to external environments, which may explain the observation of the slightly different fluorescence peak ($\lambda_F$) of compound **1** when doped in different hosts.

In the case of the $T_1$ state, both hole and particle have even contributions across the whole molecule. The virtually full overlap of hole and particle is suggestive of the enhanced locally excited (LE) character in $T_1$ with respect to $S_1$. The observation of the vibronic structure in the emission spectrum of **1** in frozen tetrahydrofuran at 77 K is in support of the LE character (Supplementary Fig. 20). This LE character results in a challenging transition from $T_1$ to $S_0$. This is supported by the small $k_{p,r}$ and $k_{p,nr}$ of 1@PA6 (0.32 and 0.34 s$^{-1}$, respectively), in contrast to the large $k_{isc}$ in the order of 10$^7$ s$^{-1}$ (Table 1). Furthermore, using the spin-orbit mean-field (SOME) methodology based on the excited-state wave functions derived from time-dependent density functional theory (TDDFT) computations, the SOC matrix elements ($\xi$) were calculated (Fig. 5b). Compound **1** has a relatively small $\xi(S_0\text{-}T_1)$ value of 0.05 cm$^{-1}$, which will retard the transition from $T_1$ to $S_0$ and thus elongates the phosphorescence lifetime. The $\xi(S_1\text{-}T_1)$ value is calculated to be 0.03 cm$^{-1}$. Considering the low $\xi(S_1\text{-}T_1)$ value and the relatively high energy gap between $S_1$ and $T_1$ (1.12 eV), the ISC process may occur primarily toward higher triplet states, e. g. $T_4$ and $T_8$ with larger SOC constants and smaller energy gaps.

### Application studies of 1@PA6

PA6 has a thermal degradation temperature of over 380 °C and a melting temperature of around 220 °C (Supplementary Fig. 42). The doping of PA6 with **1** has little impact on these properties. The good thermal properties make nylons appealing for practical applications. Nylons are used as fiber materials in many aspects. Based on the melting preparation method and the ductility of PA6, the 1@PA6 composite can be easily processed into 1D microfibers by dipping a glass rod into the composite melt and then pulling it out gently. These fibers have a diameter of 100 – 200 μm and a smooth surface. They retain the blue fluorescence under UV excitation with $\tau_F$ of 4.1 ns and $\Phi_F$ of 45.0% and persistent RTP with $\tau_P$ of 1.04 s and $\Phi_P$ of 16.2%,

respectively (Fig. 6a and Supplementary Fig. 43 and Supplementary Movie 2). The elastic strength of the 1@PA6 fiber is similar to that of pure PA6 fiber prepared by the same procedure, as is demonstrated by the stress-strain test (Fig. 6e).

Considering PA6 has an ordered molecular arrangement, we hypothesize that the luminescent 1@PA6 fiber may transmit photons as an active optical waveguide[43–47]. Using a piece of fiber of 1@PA6 with 150 μm in diameter and 1.0 mm in length as an example, we examined the optical waveguide property by performing spatially resolved luminescence measurements with a 405 nm excitation laser (Supplementary Fig. 44). The obtained results indicate that the 1@PA6 fiber exhibits both blue fluorescence and green phosphorescence waveguiding. Upon the application of the laser excitation, it behaves as a blue fluorescence waveguide with an optical loss coefficient ($\alpha$) of 2.6 dB·cm$^{-1}$ at 460 nm (Fig. 6b–d), as determined by fitting the function of log ($I_{tip}/I_{ex}$) = -$\alpha d$, where $I_{tip}$ and $I_{ex}$ represent the emission intensities at the emitting tip and the excited site, respectively, and $d$ refers to the light propagation distance between them. After turning off the excitation source, the fiber acts as a green phosphorescence waveguide with a small $\alpha$ of 0.005 dB·cm$^{-1}$ at the emission maximum of 520 nm (Fig. 6f–h). The results reveal that the generated photonic signals (both fluorescence and afterglow emissions) are efficiently restricted and transferred along the 1D fiber with no significant optical signal loss. This represents one scarce example of organic material that exhibits dual-emission waveguide property[48]. In addition, compared to known organic materials that commonly show waveguide behavior in μm range[49,50], the demonstration of 1@PA6 fiber for mm-range active waveguide represents a significant advance in the field of organic photonics.

Furthermore, the 1@PA6 melt can be easily processed into afterglow objects with various shapes using a melt casting technique on iron molds (Fig. 7a). In addition, because of the high flexibility and mechanical stability of nylon microfibers, the above-prepared 1@PA6 fiber can be woven into a nylon bracelet showing blue fluorescence and green afterglow luminescence (Fig. 7b). Taking advantage of the distinct time-resolved luminescent feature of the RTP nylons doping with different luminophores, a dual information encryption and

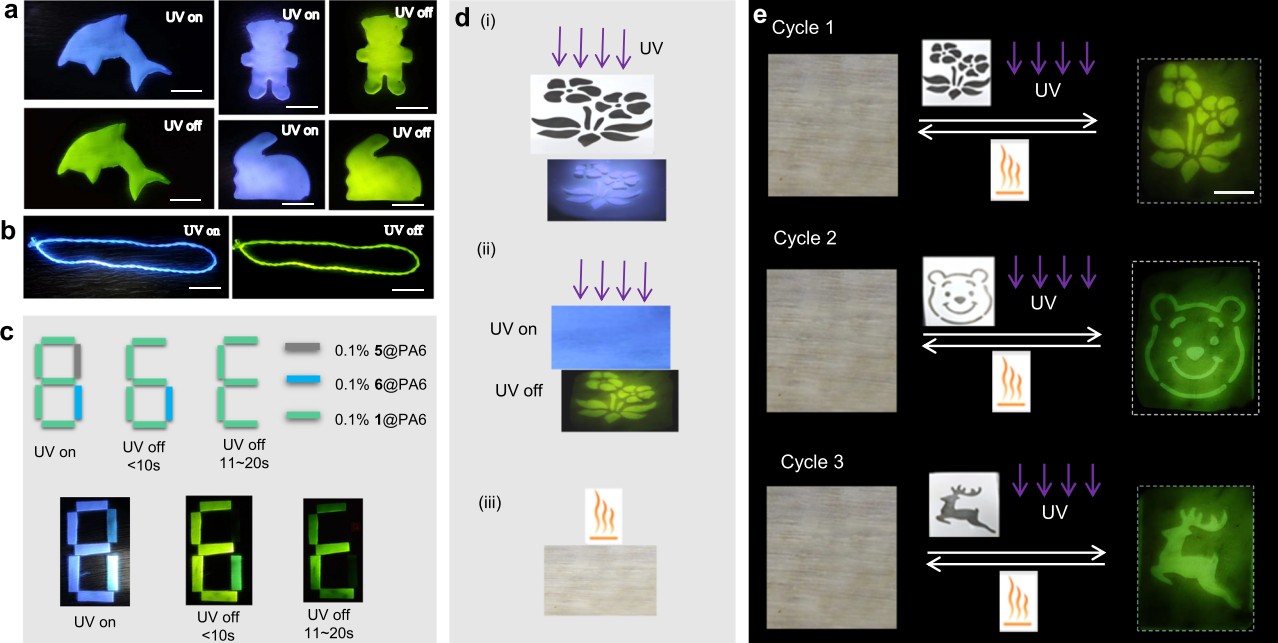

**Fig. 7 | Description of 1@PA6 film for anti-counterfeiting and afterglow display. a** Various fluorescent and afterglow-luminescent patterns prepared from a melted mixture of **1** (0.1%) and PA6. **b** A bracelet is woven from **1**@PA6 fibers showing blue fluorescence and green afterglow light. **c** Diagram depicting the dual information encryption and decryption of the signal of 'E' by using a combination of **1**@PA6, **5**@PA6, and **6**@PA6 melts. **d** Overview of the procedure of (i) writing by UV irradiation through a mask, (ii) reading of afterglow labeling, and (iii) erasing by heating on a rewritable film of **1**@PA6. **e** Demonstration of three-cycle reusable afterglow labeling via masked photo printing technology. All pictures were taken using a cell phone. The scale bar is 2 cm for all images.

decryption technology is demonstrated. We choose PA6 films doped with **1**, **5**, and **6** as the materials for information storage because they exhibit the same blue fluorescence when excited by a 365 nm UV lamp but different afterglow durations when the excitation is turned off. As illustrated in Fig. 7c, the true signal of 'E' can be encrypted beneath the bogus information of '8' under UV irradiation using two continuous encryption processes. The different strokes of this signal are made of **1**@PA6, **5**@PA6, and **6**@PA6 melts, as shown in Fig. 7c. The correct way to obtain the true information is by turning off the optical excitation and reading the signal after waiting for at least 10 s. The signal appears as another fake information of '6' when it is read immediately (within 10 s) after turning off the light. This additional layer of protection significantly improves the overall information security. As a result, these time-resolved RTP materials represent excellent candidates for information encryption applications.

The **1**@PA6 film is further chosen as a rewritable recording medium to attain programmable and reusable lifetime-encrypted security labeling on the basis of the reversible photo-responsive afterglow. As a demonstration, a large-area afterglow film (5 cm × 8 cm) is prepared by melt casting, followed by cooling to room temperature (Fig. 7d). This film is covered with a mask with a specific hollowed-out pattern. The photoexcited afterglow label can then be easily obtained through an ink-free UV-writing technology, and this provisionally stored label characterized by high-contrast afterglow emission can be conveniently recognized by the naked eye after the excitation source is removed. After heating at 100 °C for 5 s, the afterglow emission disappears immediately, and the printed label is erased. This printing-reading-erasing cycle can be repeated using masks with different patterns. As shown in Fig. 7d, three such cycles are carried out to present high-resolution afterglow images with a flower, bear, and deer pattern, respectively, demonstrating the reusable property of the recording medium (Fig. 7e). Therefore, by combining photoexcitation and thermal deactivation of the afterglow emission, these films can be used as rewritable

recording medium with environmentally friendly printing and erasing operations.

## Discussion

In summary, a series of cyano-substituted benzimidazole derivatives are synthesized and doped into nylon matrices to produce long-lived and brightly luminescent RTP polymers. Impressively, the doped materials exhibit long phosphorescence lifetimes of up to 1.5 s and a high phosphorescence quantum efficiency of 48.3% at the same time. Based on the studies of control compounds and different polymer matrices, the high RTP efficiency is the synergistic effect of several factors. The host-guest hydrogen bonding interaction allows guest molecules to be effectively dispersed into the host material. This facilitates potential energy transfer between host and guest, boosting the generation of triplet excitons from guest molecules. Nylons with short alkyl chains provide a rigid and compact environment to protect the generated triplet excitons from being quenched by oxygen, leading to excellent RTP overall performance. Owing to the processability of nylons and the homogeneous distribution of dopants in the matrix, these materials can be fabricated as 1D microfibers with dual fluorescence and phosphorescence waveguide capability in the mm range. Meanwhile, these doped polymers demonstrate intriguing potential applications such as afterglow displays and multilevel information encryption. This work not only presents a class of advanced afterglowing materials with great potential for photonic and information-related applications but also illustrates a feasible strategy for the fabrication of RTP materials by doping rigid polymers with luminophores.

### Methods reagents and materials

Unless otherwise stated, all reagents used in the experiments are purchased from commercial sources without further purification. Nylons are purchased from Shanghai Macklin Bio-chemical Technology Co., Ltd. The syntheses of luminophores are provided in the Supplementary Materials.

## Physical measurements

All data of PL spectra, lifetimes ($\tau_F$ and $\tau_P$) and quantum yields ($\Phi_F$ and $\Phi_P$) are recorded on Edinburgh FLS 980 fluorescence spectrophotometer. Stead-state PL spectra are obtained with a Xe lamp as the excitation source. Prompt and delayed PL spectra are obtained with a microsecond flash lamp (100 W) without or with a delay period. $\tau_F$ is recorded with a picosecond pulsed LED (320 or 360 nm). $\tau_P$ is recorded with a xenon lamp (450 W) or microsecond flash lamp when it is longer or shorter than 1 s, respectively. The absolute emission quantum yields are measured in air with an integrating sphere but tested separately with a Xe lamp for $\Phi_F$ and a microsecond flash lamp (100 W) for $\Phi_P$ as the excitation source, respectively. During the measurement of $\Phi_P$, a time delay of 1 ms is applied to eliminate the influence of fluorescence. FTIR spectra are carried out using a Bruker VERTEX 70 v. TGA is recorded on Netzsch STA449F3 in nitrogen atmosphere from 20 to 600 °C with a ramping rate of 10 °C min-1. PXRD spectra are performed on the Rigaku D/max-2500 instrument (Cu Kα, 1.54 Å). SEM images are obtained by a Hi-tachi SU8010 scanning electron microscope. Photographs and movies of steady-state photoluminescence and afterglow are taken by a Smartphone camera.

## Preparations of doped films

Doped nylon samples were obtained by grinding a mixture of nylon and luminophore at an indicated doping ratio of the luminophore (0.1 wt% unless otherwise stated), followed by melting at a suitable temperature (PA6: 180 °C; PA6/6: 210 °C; PA6/10: 190 °C; PA12: 160 °C) and subsequent fast (in 10 s) or slow natural cooling (annealing in 30 min) to rt. The doped polystyrene samples were prepared by the same melting method at 150 °C, followed by fast cooling to rt. The doped PVA samples were prepared by dispersing the luminophore into the aqueous solution of PVA at 90 °C, followed by drop-casting onto the quartz substrate and a post-thermal annealing treatment at 80 °C for 30 min to remove the water residue. The doped PMMA samples were prepared by dissolving the luminophore into the solution of PMMA in tetrahydrofuran at 60 °C, followed by drop-casting onto the quartz substrate and a post-thermal annealing treatment at 60 °C for 30 min to remove the tetrahydrofuran residue. If unless otherwise noted, these doped films were tested under the same air conditions, with a guest molecule concentration of 0.1 wt% for all samples and the same amount of sample for each measurement.

## Preparations of microfibers and optical waveguide measurements

The microfibers of 0.1 wt% **1**@PA6 were prepared by dipping a glass rod into the composite melt at 180 °C, followed by pulling the rod out gently at a speed of around 10 cm/s. The waveguide measurements were performed according to the experimental step shown in Supplementary Fig. 44. The emission was collected from the tip of the fiber by the objective with a back-scattering configuration and analyzed by the spectrometer with a CCD. The fluorescence and phosphorescence waveguiding were measured separately. The fluorescence waveguiding was measured by directly recording the emission signal upon exciting with a CW laser. The phosphorescence waveguiding was measured by recording the emission after immediately blocking the excitation laser light.

## Theoretical calculation

Unless otherwise noted, DFT calculations were performed using Gaussian 09 program[51]. The ground state ($S_0$) structure was optimized with B3LYP/6-311 g(d,p)[52,53]. TDDFT calculations were performed on the same level of theory. The electron transition characterization and natural transition orbitals were obtained by electron excitation analysis performed using the Multiwfn program from the transition density matrix of TDDFT calculations[54]. The SOC matrix elements between excited states were predicted based on the single crystal structure (the positions of hydrogen atoms are optimized by DFT calculations; Supplementary Data 1) by ORCA 5.0.1 program using the ωB97X-D3 functional and the def2-SV(P) basis set and def2/J auxiliary basis set[55].

## Data availability

All data needed to evaluate the conclusions in the paper are present in the paper and/or the Supplementary Materials. Additional data related to this paper may be requested from the authors. The X-ray crystallographic coordinates for structures reported in this study have been deposited at the Cambridge Crystallographic Data Center (CCDC), under deposition numbers CCDC 2337657 for compound **1** and CCDC 2337687 for compound **2**. These data can be obtained free of charge from The Cambridge Crystallographic Data Center via www.ccdc.cam.ac.uk/data_request/cif.

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

## Acknowledgements

This work is financially supported by the National Natural Science Foundation of China (grants 21925112 Y.-W.Z., 22305251 Z.-Q.L., and 22090021 Z.-L.G.) and National Key R&D Program of China (grants 2023YFE0125200 Y.-W.Z. and 2022YFA1204401 Z.-L.G.). Z.-Q.L.

acknowledges the support of BMS Junior Fellow of BNLMS. J.-Y.S. is grateful for the support of the Youth Innovation Promotion Association CAS.

## Author contributions

Conceptualization: D.-X.M., Z.-L.G., and Y.-W.Z. Methodology: D.-X.M., J.-Y.S., and Z.-Q.L. Investigation: D.-X.M., Z.-Q.L., and K.T. Supervision: Y.-W.Z. Writing-original draft: D.-X.M. Writing-review and editing: Y.-W.Z.

## Competing interests

The authors declare no competing interests.
