## [Peer Review File · Nature Communications]

Nylons with Highly-Bright and Ultralong Organic Room-Temperature PhosphorescenceREVIEWER COMMENTS

Reviewer #1 (Remarks to the Author):

The authors present six different imidazole derivatives as new organic emitters that exhibit efficient room-temperature phosphorescence (RTP) when dispersed in nylons as polymeric hosts. These organic RTP emitters have numerous potential applications, including waveguiding, anti-counterfeiting, and afterglow displays, as demonstrated in this study. The authors investigated the characteristics of RTP properties of different imidazole derivatives doped into nylon-6, PMMA, and PVA matrices. They achieved remarkably high phosphorescence quantum yields (up to 48.3%) and almost 1.5 s of lifetime of afterglow when nylon was used as the host. Considering that organic RTP emitters with even higher phosphorescence quantum yields are known, the scientific novelty of this study mainly contributes to the understanding of the nature of RTP efficiency of guest-host co-films, which is still scarcely studied. However, it is unclear why the developed compounds showed efficient RTP in nylons but poor RTP in other polymers such as PMMA and PVA.

Other comments

1. The authors initially believed that the efficient formation of hydrogen bonds caused the RTP of compounds in nylons. However, later, it looks like they refuted this assumption, claiming only that a „strong polymer-luminophore interaction is important to obtaining high RTP performance“.

For example, page 6. “ The long lived RTP phenomenon of doped polymers could be partially ascribed to the rigid environment surrounding the luminophore molecules, which also effectively shields these luminophores from the quenching of excitons by oxygen and water. The hydrogen bonding networks formed among nylon chains and between nylons and luminophores efficiently restrict the molecular motion of luminophores and thus suppress the nonradiative decay of triplet excitons. These factors are believed to be critical for the observation of the long persistent and bright RTP of PA6 doped with 1 4 and PA6/6 and PA6/10 doped with 1”

As discussed below, the PMMA- and PVA-based host-guest films were also studied, but their phosphorescence lifetimes and, most importantly, phosphorescence quantum yields were below the tested compound1-6PA film. The abovementioned polymers also confine a luminophore by forming multiple intermolecular interactions. It is worth putting more stress on the compatibility of polymer crystalline lattice size/ size of the guest molecule as a critical point resulting in the superior RTP performance of compound1-6PA film. Why are polymer-luminophore interactions not strong enough when PMMA and PVA were used as hosts? Why the developed compounds showed efficient RTP in nylons but poor RTP in other polymers such as PMMA and PVA?

2. The radiative/nonradiative processes of the molecular systems based on newly developed RTP emitters and nylon matrix should be investigated. For example, singlet-triplet intersystem crossing (ISCs) efficiencies should be added to Table 1 and discussed in the main text. Why does the fluorescence of

compound 1 peak at different wavelengths when doped in different hosts?

3. How was the prompt spectrum in Fig 1b recorded? For the studied compounds, not only prompt and delayed spectra but also steady-stated PL spectra should be recorded at different conditions (e.g. oxygen/air/vacuum). Verifying if 1@PA6 displays white emission under UV excitation would be interesting since the FL and RTP have similar QYs.

4. In the introduction, few organic RTP emitters are mentioned, reaching RTP QYs of 14.7%. However, there are organic RTP materials with higher values of RTP QYs in the literature (e.g. DOI: 10.1039/D3TC04514E; 10.1021/acssuschemeng.3c04011; 10.1016/j.snb.2022.132727).

5. The conditions of experiments are not clear. Were the samples based on compound 1 dispersed in nylons, PMMA, or PVA tested at the same conditions? Was the concentration of the guests the same? Were the thicknesses of the samples the same? Were the samples tested in air or an inert atmosphere? The section "Methods" should be revised to provide additional details on PLQY measurements. Were the PLQYs measured in the air? How were QYs of fluorescence and phosphorescence separately measured?

6. Page 6: "These results suggest that the cyano groups of 1 are crucial in maintaining the high RTP performance 1@PA6, while the imidazole N H unit is not indispensable. Fourier transform infrared (FTIR) spectroscopy was subsequently performed to examine the hydrogen bonding interactions in 1@PA6 film (Fig. 3d). In particular, the stretching vibration signal of the cyano group shifts from 2237 cm⁻¹ for the pure 1 solid to 2229 cm⁻¹ for 1@PA6 film. The change is ascribed to forming a relatively rigid hydrogen bonding network between the amide protons of nylon 6 and the CN groups of 1, contributing to the significant phosphorescence enhancement in 1@PA6."

It would be interesting to see the FTIR spectroscopy for other films, especially, for the film of compound 2-6pa, that has cyanogroups in ortho-position, to see if the hydrogen bondings are formed in this case. It would additionally shed light on what is crucial in the RTP of studied imidazole derivatives-pa6 films: the hydrogen bonding provided by cyano groups or compatibility of polymer lattice size/ size of the molecule of the guest luminophore?

7. Figures 1b, 2b Figures S17, S18, S22. All the guest compounds are very similar in structure. Why does compound 1-6PA guest-host solid mixture exhibit an unstructured RTP profile, while the RTP profiles of other films are structured, as well as the RTP profiles of pure imidazole derivatives in frozen THF? A possible explanation of this observation could be added.

8. If Fig3b shows "Schematic illustration of hydrogen bonding networks among γ -phase nylon 6 polymers.", it should be clarified how values of "a", "b", and "c" were obtained.

Reviewer #2 (Remarks to the Author):

Over the past decades, organic room-temperature phosphorescence (RTP) has evolved as one of the hottest topics in multidisciplinary fields of optoelectronic devices, optical imaging, sensors, photocatalysis, biotherapy, and information security, among others. Compared with crystalline counterparts, amorphous polymer RTP materials show greater potential due to their good processability and flexibility. In this manuscript, Zhong et al. reported a type of highly-efficient ultralong RTP nylons via doping cyano-substituted benzimidazole derivatives into the nylon matrix PA6. When the dopant is the luminophore 2-phenyl-1H-4,7-dicyanobenzo[d]imidazole (1), the homogeneously doped nylons (1@ γ -PA6) exhibited RTP property with a lifetime of up to 1.5 s and quantum efficiency of 48.3%. Authors claimed that rich hydrogen bonding interaction between luminophores and nylon is the key factor for achieving highly-efficient ultralong RTP, by combining experimental and theoretical results. Most importantly, they further fabricated a kind of 1D active optical waveguide which transmits blue fluorescence and green phosphorescence signals across the millimeter distance. They also showcased how RTP-doped nylons are utilized for various purposes such as information encryption and rewritable recording. In all, this paper is well constructed, and the authors presented coherent arguments that were supported by the experiments. Therefore, I would recommend this manuscript for publication on Nature Communications after major revision. The following comments may help improve the manuscript.

1. Authors should cite some important references about 1D phosphorescent waveguides, such as *Angew. Chem. Int. Ed.* 2023, 62, e202309913, *Angew. Chem. Int. Ed.* 2022, 61, e202208735, *Adv. Mater.* 2021, 33, 2007571, *Angew. Chem. Int. Ed.* 2019, 58, 15128–15135.

2. Fitting curves of decay profiles should be provided, such as Figure S16 and Figure S21. Moreover, the lifetime 4 ns in the sentence “it has a short lifetime of 4.0 ns (τ_F), suggestive of a typical fluorescence behavior (Fig. S16)” is not consistent with the legend 3.95 ns written in Figure S16.

3. On page 3, I would suggest authors first discuss PXRD data of γ -phase and α -phase nylons, then present the photophysical properties of the doped materials based on the γ -phase nylons, which would provide a clearer structure-property relationship.

4. The authors have tested nylon 6/6, nylon 6/10 and nylon 12 as the matrix for the control experiments. Therefore, the chemical structures of nylon 6/6, nylon 6/10 and nylon 12 should be provided.

5. Figure 2e is confusing, in which authors used the same green color for both lifetime and efficiency legends. Please double check.

6. Authors claimed that stretching vibration signal of the cyano group shifts from 2237 cm^{-1} for the pure 1 solid to 2229 cm^{-1} for 1@PA6 film. However, the shifting direction for the pure 1 and 1@PA6 film from 2229 to 2237 cm^{-1} in Figure 3d is NOT consistent with the corresponding magnified inset.

7. The FT-IR spectra present intense N-H stretching vibration in 1@PA6 film. Authors should also consider the contribution of N-H to hydrogen bonding for enhancing RTP performance.

8. Authors should provide the preparation details about the 1D microfibers such as the processing

temperatures and time. Moreover, the authors need to add the DSC data to show the glass transition and/or melting temperatures of 1@PA6.

9. Most of the images in Figure 6a are pure black without any shapes. Please check.

10. Since the fluorescence spectrum overlaps with the RTP spectrum, it is very important to provide detailed procedures about how to distinguish RTP waveguide from the fluorescence waveguide.

Reviewer #3 (Remarks to the Author):

Metal-free polymer materials exhibiting room-temperature phosphorescence (RTP) are good candidates for practical applications, as they show the great advantages of flexibility, easy to process, high modifiability, and good biocompatibility. This manuscript reports a rational strategy to achieve persistent RTP by embedding cyano-substituted benzimidazole derivatives into nylon matrix. These materials show ultralong phosphorescence lifetimes and high phosphorescence quantum efficiency, and perform well in dual information encryption and rewritable recording. However, the authors must provide more convincing evidence before boasting the achievement of such a result. Hence, this manuscript could be published in this journal, but major revision is required.

1. The fluorescence and phosphorescence properties of pure PA6 at 77k and room temperature should be investigated. Besides, the fluorescence and phosphorescence properties of crystals 1, 2, 3, and 4 at room temperature should also be provided.

2. The authors should thoroughly discuss the reasons why 1@PA12 exhibits poor phosphorescence properties compared to 1@PA6, 1@PA6/6, and 1@PA6/10.

3. The authors emphasize that the rich hydrogen bonding interaction between luminophores and nylon is crucial for achieving the high-performance RTP. Polyvinyl alcohol (PVA) is considered to be a rigid polymer matrix with hydrogen bonding networks. While, compared with 1@PA6, 1@PVA possesses comparable phosphorescence lifetime but extremely lower phosphorescence efficiency. The authors should explain the reason. Does energy transfer between host and guest molecules or clustering-triggered emission (CTE) exist in 1@PA6?

4. In the part of theoretical calculations, the functional and basis set used for ground state geometries optimizations as well as the program used for DFT calculation are not stated. Also, necessary references are required.

5. The signal intensity of NMR spectra is too weak (Figure S3, S5, S6, S9, S12). Signal splitting in ¹H NMR is not clear (Figure S11). NMR spectra with higher quality are required.

6. There are two Fig. S20 in the Supplementary Information, the authors should reorder the figures in the Supplementary Information.

7. The manuscript contains a few typos, such as, “a-phase nylons” should be “ α -phase nylons”, “ τ F (ns)c” in Table 1.

Reviewer #4 (Remarks to the Author):

Responses to the Reviewers comments

Reviewer #1:

The authors present six different imidazole derivatives as new organic emitters that exhibit efficient room-temperature phosphorescence (RTP) when dispersed in nylons as polymeric hosts. These organic RTP emitters have numerous potential applications, including waveguiding, anti-counterfeiting, and afterglow displays, as demonstrated in this study. The authors investigated the characteristics of RTP properties of different imidazole derivatives doped into nylon-6, PMMA, and PVA matrices. They achieved remarkably high phosphorescence quantum yields (up to 48.3%) and almost 1.5 s of lifetime of afterglow when nylon was used as the host. Considering that organic RTP emitters with even higher phosphorescence quantum yields are known, the scientific novelty of this study mainly contributes to the understanding of the nature of RTP efficiency of guest-host co-films, which is still scarcely studied. However, it is unclear why the developed compounds showed efficient RTP in nylons but poor RTP in other polymers such as PMMA and PVA.

1. The authors initially believed that the efficient formation of hydrogen bonds caused the RTP of compounds in nylons. However, later, it looks like they refuted this assumption, claiming only that a “strong polymer-luminophore interaction is important to obtaining high RTP performance”.

For example, page 6. “The long lived RTP phenomenon of doped polymers could be partially ascribed to the rigid environment surrounding the luminophore molecules, which also effectively shields these luminophores from the quenching of excitons by oxygen and water. The hydrogen bonding networks formed among nylon chains and between nylons and luminophores efficiently restrict the molecular motion of luminophores and thus suppress the nonradiative decay of triplet excitons. These factors are believed to be critical for the observation of the long persistent and bright RTP of PA6 doped with 1 4 and PA6/6 and PA6/10 doped with 1”

As discussed below, the PMMA- and PVA-based host-guest films were also studied, but their phosphorescence lifetimes and, most importantly, phosphorescence quantum yields were below the tested compound 1-PA6 film. The abovementioned polymers also confine a luminophore by forming multiple intermolecular interactions. It is worth putting more stress on the compatibility of polymer crystalline lattice size/ size of the guest molecule as a critical point resulting in the superior RTP performance of compound 1-PA6 film. Why are polymer-luminophore interactions not strong enough when PMMA and PVA were used as hosts? Why the developed compounds showed efficient RTP in nylons but poor RTP in other polymers such as PMMA and PVA?

Response: Thanks for the comments. Following your suggestions and those from other reviewers, we have included some additional characterizations on the rtp properties of these doped polymers. The excellent rtp properties of the 1@PA6 polymer are now ascribed to the synergistic effect of the homogeneous distribution of guest molecules by hydrogen bonding interactions, the rigid environment of PA6, and

the energy transfer between host and guest molecules. The hydrogen bonding interaction alone is certainly not enough to induce the desired rtp properties, as supported by the poor rtp performances of **1**@PVA and **1**@PA12 polymers and the crystalline sample of **1**, all of which are rich in hydrogen bonding interactions. The PVA, PA12, and PMMA polymers are relatively flexible due to the presence of long alkyl chains. They are not good energy donors due to their poor phosphorescent properties. These factors are the critical reasons accounting for the poor rtp performances when they are used as the host for **1**. Regarding the compatibility of polymer crystalline lattice size versus the size of the guest molecule, we tend to believe that this is a minor issue in the current system, considering that α - and γ -PA6 with different crystalline lattices are both excellent hosts and the sizes of the small molecular luminophores **1** – **7** do not differ significantly. Corresponding discussions have been updated as follows in the revised manuscript.

Considering that the rich amide groups of nylon polymers may assist the occurrence of clustering-triggered emission and phosphorescence,^{13,22} we examined the luminescence properties of the host materials. Interestingly, the pure PA6 film exhibits weak fluorescence and phosphorescence dual luminescence with $\lambda_F = 350$ nm, $\Phi_F = 0.4\%$, $\lambda_P = 450$ nm, $\Phi_P = 0.5\%$ at rt (Fig. 4c and S37). The τ_F at 350 nm is 1.30 ns, which remains essentially unchanged after doping with compound **1** (Fig. 4d). The τ_P of PA6 is wavelength-dependent and it exhibits distinct changes after doping with **1**. In the presence of **1**, the τ_P at 400 nm is shortened from 41.52 to 21.47 ms and that at 420 nm is shortened from 51.27 to 22.17 ms, respectively (Fig. 4e and 4f). This suggests that triplet-to-triplet Dexter-type energy transfer occurs from the PA6 host to **1**, which is also supported by the presence of UV absorption of PA6 and the energy level match between PA6 and **1** (Fig. 4b and 4c). The τ_P at 450 nm of PA6 becomes longer after doping with **1**, which should be caused by the influence of the long-lived phosphorescence of **1**. In addition to PA6, weak RTPs are observed for PA66 and PA610 and similar changes of τ_P occur in the presence of **1** (Fig. S38). These results support that the presence of host-guest energy transfer contributes to the excellent RTP performance of **1**@PA6, **1**@PA6/6, and **1**@PA6/10. In contrast, essentially no distinct RTP could be observed from the flexible polymer PA12, PVA, and PMMA. In addition, though weak fluorescence and RTP are observed for PS, no potential energy transfer is evidenced in **1**@PS (Fig. S39). The lack of efficient energy transfer in these flexible polymer composites is believed to be accountable for their poor RTP performance.

Fig. 4 A comparison study of PA6, **1**, and 0.1 wt% **1@PA6**. **a** FTIR spectra. **b** Absorption spectra of PA6 and **1@PA6** films and the solution of **1** in THF. **c** Steady-state and delayed phosphorescence spectra of PA6 and **1@PA6**. **d** Fluorescence lifetime decays and fitted curves at 350 nm of PA6 and **1@PA6**. **e,f** Phosphorescence lifetime decays and fitted curves at indicated wavelength of (e) PA6 and (f) **1@PA6** (excited at 290 nm).

2. The radiative/nonradiative processes of the molecular systems based on newly developed RTP emitters and nylon matrix should be investigated. For example, singlet-triplet intersystem crossing (ISCs) efficiencies should be added to Table 1 and discussed in the main text. Why does the fluorescence of compound **1** peak at different wavelengths when doped in different hosts?

Response: the information on the radiative and nonradiative rate constants and ISC rate constant and efficiency are now provided in Table 1 and proper discussions have been provided in the revised manuscript.

Compound **1** is characterized with a hybrid local and charge transfer (HLCT) excited state due to the strong electron-withdrawing effect of the cyano group. This type of luminescence is known to be sensitive to external environments. The observation of the different fluorescence peak of compound **1** when doped in different hosts is likely due to the charge-transfer luminescence characteristics. Corresponding revision has been added in the “Theoretical Calculation and Discussion on Photophysical Properties” section of the revised manuscript.

3. How was the prompt spectrum in Fig 1b recorded? For the studied compounds, not only prompt and delayed spectra but also steady-stated PL spectra should be recorded at different conditions (e.g. oxygen/air/vacuum). Verifying if **1@PA6** displays white emission under UV excitation would be interesting since the FL and RTP have similar QYs.

Response: The fluorescence spectrum in Fig. 1b is the steady-state spectrum measured in air. Correction has been made to the figure. The prompt, delayed, and steady-stated

PL spectra have been measured at different conditions (oxygen/air/vacuum) (Figure S17). The measurement equipment information is now clearly given in the figure caption and the Methods section of the main article. The shapes of these spectra are independent on the measurement condition. The doped polymer does not display white emission under UV excitation, probably due to the significant difference in the luminescence and phosphorescence lifetime. In addition, the phosphorescence lifetime of the 1@PA6 film is almost the same under different measurement conditions, suggesting the excellent oxygen-shielding effect of the polymer matrix. Corresponding data and discussion have been updated in the revised manuscript.

Figure S17. (a) Steady-state, (b) prompt, and (c) delayed photoluminescence spectra and (d) lifetime decay curves of the phosphorescence emission band at 510 nm of 0.1 wt% 1@PA6 measured in air, under vacuum, or after being exposed to oxygen for 30 min. The steady-state emissions are obtained with a xenon lamp (450 W). The prompt and delayed emissions are obtained with a microsecond flash lamp (100 W) without or with time delay (5 ms), respectively.

4. In the introduction, few organic RTP emitters are mentioned, reaching RTP QYs of 14.7%. However, there are organic RTP materials with higher values of RTP QYs in the literature (e.g. DOI: 10.1039/D3TC04514E; 10.1021/acssuschemeng.3c04011; 10.1016/j.snb.2022.132727).

Response: Thanks for bringing these works into our attention. These literatures are now cited in refs 35-37 in the manuscript. Specifically, the following discussion has been updated in the manuscript.

PA6 doped with triphenylamine boric acid shows RTP with Φ_p of 14.7% and τ_p of 724 ms.³⁴ The RTP properties of these reported doped nylons are much inferior to those of the state-of-the-art RTP organic crystals and polymeric materials.^{1,21,35-37} It is of high significance and urgency to develop an excellent type of luminophore dopant that is able to achieve highly efficient RTP with nylons.

5. The conditions of experiments are not clear. Were the samples based on compound **1** dispersed in nylons, PMMA, or PVA tested at the same conditions? Was the concentration of the guests the same? Were the thicknesses of the samples the same? Were the samples tested in air or an inert atmosphere? The section “Methods“ should be revised to provide additional details on PLQY measurements. Were the PLQYs measured in the air? How were QYs of fluorescence and phosphorescence separately measured?

Response: The doped films were tested under the same air conditions, with a guest molecule concentration of 0.1 wt% for all samples and the same amount of sample for each measurement, if unless otherwise noted. This point has been mentioned in the “Methods/Preparations of doped films” section.

In addition, The “Methods/Physical measurements” section has been updated to provide all details on the measurements of different emission spectra and photoluminescence quantum yields. Specifically, the following details are provided in the revised manuscript:

All data of PL spectra, lifetimes (τ_F and τ_P) and quantum yields (Φ_F and Φ_P) are recorded on Edinburgh FLS 980 fluorescence spectrophotometer. Stead-state PL spectra are obtained with a Xe lamp as the excitation source. Prompt and delayed PL spectra are obtained with a microsecond flash lamp (100 W) without or with a delay period. τ_F is recorded with a picosecond pulsed LED (320 or 360 nm). τ_P is recorded with a xenon lamp (450 W) or microsecond flash lamp when it is longer or shorter than 1 s, respectively. The absolute emission quantum yields are measured in air with an integrating sphere, but tested separately with a Xe lamp for Φ_F and microsecond flash lamp (100 W) for Φ_P as the excitation source, respectively. During the measurement of Φ_P , a time delay of 1 ms is applied to eliminate the influence of fluorescence.

6. Page 6: “These results suggest that the cyano groups of **1** are crucial in maintaining the high RTP performance **1**@PA6, while the imidazole N-H unit is not indispensable. Fourier transform infrared (FTIR) spectroscopy was subsequently performed to examine the hydrogen bonding interactions in **1**@PA6 film (Fig. 3d). In particular, the stretching vibration signal of the cyano group shifts from 2237 cm⁻¹ for the pure **1** solid to 2229 cm⁻¹ for **1**@PA6 film. The change is ascribed to forming a relatively rigid hydrogen bonding network between the amide protons of nylon 6 and the CN groups of **1**, contributing to the significant phosphorescence enhancement in **1**@PA6.”

It would be interesting to see the FTIR spectroscopy for other films, especially, for the film of compound **2-6PA**, that has cyano groups in ortho-position, to see if the hydrogen bondings are formed in this case. It would additionally shed light on what is crucial in the RTP of studied imidazole derivatives-pa6 films: the hydrogen bonding provided by cyano groups or compatibility of polymer lattice size/ size of the molecule of the guest luminophore?

Response: the FTIR spectra of PA6 doped with **2**, **3**, or **4** have been measured (Figure S31). According to the spectral shift of the CN vibration signal, hydrogen bondings are formed in all of these samples, including compound **2** possessing cyano groups in ortho-position. As was discussed in the reply to the first question, the hydrogen bonding alone is not sufficient enough to induce excellent RTP. The polymer crystalline lattice size is considered to be a minor factor in the current system. The efficient energy transfer between matrix and guest luminophores plays an important role in realizing RTP. Corresponding data and discussion have been updated in the revised manuscript.

Figure S31. FTIR spectra of (a) **2**@PA6, (b) **3**@PA6, and (c) **4**@PA6 in comparison with those of the powder of pure **2** – **4** and pure PA6.

7. Figures 1b, 2b Figures S17, S18, S22. All the guest compounds are very similar in structure. Why does compound 1-6PA guest-host solid mixture exhibit an unstructured RTP profile, while the RTP profiles of other films are structured, as well as the RTP profiles of pure imidazole derivatives in frozen THF? A possible explanation of this observation could be added.

Response: These compounds are characterized with a hybrid local and charge transfer (HLCT) excited state. The shape of the emission spectrum is sensitive to external environments. At 77 K in frozen THF, the non-radiative transition is almost entirely suppressed, and the observation of fine vibronic structure of the emission peak may be due to the enhanced locally excited (LE) character. This brief discussion has been added in the “Theoretical Calculation and Discussion on Photophysical Properties” section of the revised manuscript.

8. If Fig 3b shows “Schematic illustration of hydrogen bonding networks among γ -phase nylon 6 polymers.”, it should be clarified how values of “a”, “b”, and “c” were obtained.

Response: These polymer lattice size values were cited from known literatures. However, as was discussed in the in the reply to the first question, the polymer crystalline lattice size is considered to be a minor factor in the current system. The exact values of these data have been deleted in the revised manuscript.

Reviewer# 2:

Over the past decades, organic room-temperature phosphorescence (RTP) has evolved as one of the hottest topics in multidisciplinary fields of optoelectronic devices, optical imaging, sensors, photocatalysis, biotherapy, and information security, among others. Compared with crystalline counterparts, amorphous polymer RTP materials show greater potential due to their good processability and flexibility. In this manuscript, Zhong et al. reported a type of highly-efficient utralong RTP nylons via doping cyano-substituted benzimidazole derivatives into the nylon matrix PA6. When the dopant is the luminophore 2-phenyl-1H-4,7-dicyanobenzo[d]imidazole (1), the homogeneously doped nylons (1@ γ -PA6) exhibited RTP property with a lifetime of up to 1.5 s and quantum efficiency of 48.3%. Authors claimed that rich hydrogen bonding interaction between luminophores and nylon is the key factor for achieving highly-efficient utralong RTP, by combining experimental and theoretical results. Most importantly, they further fabricated a kind of 1D active optical waveguide which transmits blue fluorescence and green phosphorescence signals across the millimeter distance. They also showcased how RTP-doped nylons are utilized for various purposes such as information encryption and rewritable recording. In all, this paper is well constructed, and the authors presented coherent arguments that were supported by the experiments. Therefore, I would recommend this manuscript for publication on Nature Communications after major revision. The following comments may help improve the manuscript.

1. Authors should cite some important references about 1D phosphorescent waveguides, such as Angew. Chem. Int. Ed. 2023, 62, e202309913, Angew. Chem. Int. Ed. 2022, 61, e202208735, Adv. Mater. 2021, 33, 2007571, Angew. Chem. Int. Ed. 2019, 58, 15128–15135.

Response: Thanks for bringing these works into our attention. These literatures are now cited in refs 44-47 in the revised manuscript.

2. Fitting curves of decay profiles should be provided, such as Figure S16 and Figure S21. Moreover, the lifetime 4 ns in the sentence “it has a short lifetime of 4.0 ns (τ_F), suggestive of a typical fluorescence behavior (Fig. S16)” is not consistent with the legend 3.95 ns written in Figure S16.

Response: Fitting curves of the decay profiles in mentioned Figures have been added. A consistent lifetime of 4.0 ns is now given in Fig. S16 and the main article.

3. On page 3, I would suggest authors first discuss PXRD data of γ -phase and α -phase nylons, then present the photophysical properties of the doped materials based on the γ -phase nylons, which would provide a clearer structure-property relationship.

Response: thanks for your suggestion. The article focuses on the doping system's photophysical properties. Considering that γ -phase and α -phase nylons gave comparable results, we'd like to keep the order of presentation by first discussing the photophysical properties before the PXRD data.

4. The authors have tested nylon 6/6, nylon 6/10 and nylon 12 as the matrix for the control experiments. Therefore, the chemical structures of nylon 6/6, nylon 6/10 and nylon 12 should be provided.

Response: the chemical structures of nylon 6/6, nylon 6/10, and nylon 12 have been added in the supplementary materials (Fig. S25).

Figure S25. The chemical structures of PA6/6, PA6/10 and PA12.

5. Figure 2e is confusing, in which authors used the same green color for both lifetime and efficiency legends. Please double check.

Response: Figure 2e has been updated with clear legends.

Fig. 2e A comparison of measured phosphorescence (Phos.) lifetime and efficiency.

6. Authors claimed that stretching vibration signal of the cyano group shifts from 2237 cm^{-1} for the pure 1 solid to 2229 cm^{-1} for 1@PA6 film. However, the shifting

direction for the pure **1** and **1@PA6** film from 2229 to 2237 cm⁻¹ in Figure 3d is NOT consistent with the corresponding magnified inset.

Response: Thanks for pointing out this mistake. Correction has been made and updated in the revised manuscript.

7. The FT-IR spectra present intense N-H stretching vibration in **1@PA6** film. Authors should also consider the contribution of N-H to hydrogen bonding for enhancing RTP performance.

Response: Both Nylon 6 and **1** contain N-H bonds. In **1@PA6** film, the N-H vibration of the small amount of **1** is embedded in that of PA6. The change of the N-H vibration signal cannot be directly monitored in FTIR spectra. However, we synthesized compound **6** with a methyl substituent in replace of the N-H bond. The composite of **6@PA6** also shows high RTP performance. On the basis of this fact, the contribution of N-H to hydrogen bonding may be insignificant. This brief discussion has been mentioned in the revised manuscript.

8. Authors should provide the preparation details about the 1D microfibers such as the processing temperatures and time. Moreover, the authors need to add the DSC data to show the glass transition and/or melting temperatures of **1@PA6**.

Response: The microfibers of 0.1 wt% **1@PA6** were prepared by dipping a glass rod into the composite melt at 180 °C, followed by pulling the rod out gently in a speed of around 10 cm/s. This detail has been added to the “Methods/Preparations of Microfibers and Optical waveguide measurements” section of the revised manuscript.

The DSC data has been added in Fig. S40, which showed a melting temperature of 220 °C upon heating of the **1@PA6** film. Related data and discussion have been added in the revised manuscript.

9. Most of the images in Figure 6a are pure black without any shapes. Please check.

Response: this was likely caused by an issue of display. The figure was updated and re-uploaded as follows (now Figure 7a in revised manuscript).

10. Since the fluorescence spectrum overlaps with the RTP spectrum, it is very important to provide detailed procedures about how to distinguish RTP waveguide from the fluorescence waveguide.

Response: The fluorescence and phosphorescence waveguiding were measured separately. The fluorescence waveguiding was measured by directly recording the emission signal upon exciting with a CW laser. The phosphorescence waveguiding

was measured by recording the emission after immediately blocking the excitation laser light. This detail has been added to the “Methods/Preparations of Microfibers and Optical waveguide measurements” section of the revised manuscript.

Reviewer #3:

Metal-free polymer materials exhibiting room-temperature phosphorescence (RTP) are good candidates for practical applications, as they show the great advantages of flexibility, easy to process, high modifiability, and good biocompatibility. This manuscript reports a rational strategy to achieve persistent RTP by embedding cyano-substituted benzimidazole derivatives into nylon matrix. These materials show ultralong phosphorescence lifetimes and high phosphorescence quantum efficiency, and perform well in dual information encryption and rewritable recording. However, the authors must provide more convincing evidence before boasting the achievement of such a result. Hence, this manuscript could be published in this journal, but major revision is required.

1. The fluorescence and phosphorescence properties of pure PA6 at 77 K and room temperature should be investigated. Besides, the fluorescence and phosphorescence properties of crystals 1, 2, 3, and 4 at room temperature should also be provided.

Response: The photophysical properties of pure PA6 are studied at both room temperature and 77 K (Fig. S37). Specifically, the following discussion was added in the revised manuscript.

Considering that the rich amide groups of nylon polymers may assist the occurrence of clustering-triggered emission and phosphorescence,^{13,22} we examined the luminescence properties of the host materials. Interestingly, the pure PA6 film exhibits weak fluorescence and phosphorescence dual luminescence with $\lambda_F = 350$ nm, $\Phi_F = 0.4\%$, $\lambda_P = 450$ nm, $\Phi_P = 0.5\%$ at rt (Fig. 4c and S37).

Figure S37. (a,b) Normalized steady-state and delayed emission spectra of pure PA6 film at (a) 298 K and (b) 77 K. (c) Phosphorescence decay curves of PA6 at 450 nm at 298 K and 77 K.

The fluorescence and phosphorescence properties of crystals 1 - 4 are also examined as required. Specifically, the following discussion was added in the revised manuscript.

However, the presence of the hydrogen bonding alone cannot account for the

excellent RTP performance of **1**@PA6... The single crystal X-ray analysis of **1** and **2** demonstrates the involvement of the CN groups in the formation of hydrogen bonding (Fig. S32 and Table S2). However, the crystalline solids of **1** – **4** show very weak RTP with τ_p of shorter than 30 ms (Fig. S33 – S35).

Figure S33. Normalized steady-state and delayed emission spectra of the crystalline powder of (a) **1**, (b) **2**, (c) **3**, and (d) **4** under ambient conditions. Delayed time: 0.5 ms.

Figure S34. Lifetime decay curves of the fluorescence emission of the crystalline powder of (a) **1**, (b) **2**, (c) **3**, and (d) **4** under ambient conditions.

Figure S35. Lifetime decay curves of the phosphorescence emission of the crystalline powder of (a) **1**, (b) **2**, (c) **3**, and (d) **4** under ambient conditions. Measured after 0.5 ms of delay.

2. The authors should thoroughly discuss the reasons why **1@PA12** exhibits poor phosphorescence properties compared to **1@PA6**, **1@PA6/6**, and **1@PA6/10**.

Response: following the suggestion of the reviewer, we studied the photophysical properties of PA6 and those of other polymer matrices in the absence or presence of the dopant **1**. It was found that the host-guest energy transfer plays a critical role in influencing the RTP performance. Specifically, the following discussion was added in the revised manuscript.

Interestingly, the pure PA6 film exhibits weak fluorescence and phosphorescence dual luminescence with $\lambda_F = 350$ nm, $\Phi_F = 0.4\%$, $\lambda_P = 450$ nm, $\Phi_P = 0.5\%$ at rt (Fig. 4c and S37). The τ_F at 350 nm is 1.30 ns, which remains essentially unchanged after doping with compound **1** (Fig. 4d). The τ_P of PA6 is wavelength-dependent and it exhibits distinct changes after doping with **1**. In the presence of **1**, the τ_P at 400 nm is shortened from 41.52 to 21.47 ms and that at 420 nm is shortened from 51.27 to 22.17 ms, respectively (Fig. 4e and 4f). This suggests that triplet-to-triplet Dexter-type energy transfer occurs from the PA6 host to **1**, which is also supported by the presence of UV absorption of PA6 and the energy level match between PA6 and **1** (Fig. 4b and 4c). The τ_P at 450 nm of PA6 becomes longer after doping with **1**, which should be caused by the influence of the long-lived phosphorescence of **1**. In addition to PA6, weak RTPs are observed for PA6/6 and PA6/10 and similar changes of τ_P occur in the presence of **1** (Fig. S38). These results support that the presence of host-guest energy transfer contributes to the excellent RTP performance of **1@PA6**, **1@PA6/6**, and **1@PA6/10**. In contrast, essentially no distinct RTP could be observed from the flexible polymer PA12, PVA, and PMMA. In addition, though weak fluorescence and

RTP are observed for PS, no potential energy transfer is evidenced in **1**@PS (Fig. S39). The lack of efficient energy transfer in these flexible polymer composites is believed to be accountable for their poor RTP performance.

Fig. 4 A comparison study of PA6, **1**, and 0.1 wt% **1**@PA6. **a** FTIR spectra. **b** Absorption spectra of PA6 and **1**@PA6 films and the solution of **1** in THF. **c** Steady-state and delayed phosphorescence spectra of PA6 and **1**@PA6. **d** Fluorescence lifetime decays and fitted curves at 350 nm of PA6 and **1**@PA6. **e,f** Phosphorescence lifetime decays and fitted curves at indicated wavelength of (e) PA6 and (f) **1**@PA6 (excited at 290 nm).

3. The authors emphasize that the rich hydrogen bonding interaction between luminophores and nylon is crucial for achieving the high-performance RTP. Polyvinyl alcohol (PVA) is considered to be a rigid polymer matrix with hydrogen bonding networks. While, compared with **1**@PA6, **1**@PVA possesses comparable phosphorescence lifetime but extremely lower phosphorescence efficiency. The authors should explain the reason. Does energy transfer between host and guest molecules or clustering-triggered emission (CTE) exist in **1**@PA6?

Response: Thanks for the comments and suggestion. As is stated in the response to the last question, we studied the photophysical properties of PA6 and those of other polymer matrices in the absence or presence of the dopant **1**. It was found that the host-guest energy transfer play a critical in influencing the RTP performance. The lack of efficient energy transfer in flexible polymer composites, including **1**@PVA, is believed to be accountable for their poor RTP performance.

4. In the part of theoretical calculations, the functional and basis set used for ground state geometries optimizations as well as the program used for DFT calculation are not stated. Also, necessary references are required.

Response: the details on calculations have been updated with necessary references as follows.

All DFT calculations were performed using Gaussian 09 program.⁵¹ The ground state

(S_0) structure was optimized with B3LYP/6-311g(d,p).⁵² TDDFT calculations were performed on the same level of theory. The electron transition characterization and natural transition orbitals were obtained by electron excitation analysis performed using the Multiwfn program from the transition density matrix of TDDFT calculations.⁵³ The SOC matrix elements between excited states were predicted based on the single crystal structure by ORCA 5.0.1 program using the ω B97X-D3 functional and the def2-SV(P) basis set and def2/J auxiliary basis set.⁵⁴

5. The signal intensity of NMR spectra is too weak (Figure S3, S5, S6, S9, S12). Signal splitting in ^1H NMR is not clear (Figure S11). NMR spectra with higher quality are required.

Response: the NMR spectra have been updated with better resolutions. Due to the relatively poor solubility, the NMR signals of some spectra are still not very high. The single crystal X-ray data of compounds **1** and **2** are provided in the revised manuscript to further confirm their structures.

6. There are two Fig. S20 in the Supplementary Information, the authors should reorder the figures in the Supplementary Information.

Response: Thanks for pointing out this mistake. Correction has been made in the revised Supplementary Information.

7. The manuscript contains a few typos, such as, “a-phase nylons” should be “ α -phase nylons”, “ τ_{F} (ns)c” in Table 1.

Response: Thanks for pointing out this mistake. Corrections have been made as suggested.

REVIEWER COMMENTS

Reviewer #1 (Remarks to the Author):

The authors addressed the raised concerns. The amended manuscript is appropriate for publishing.

Reviewer #2 (Remarks to the Author):

The issues have been addressed properly. I recommend publication of this paper on this journal

Reviewer #3 (Remarks to the Author):

I am pleased to see that the authors have provided additional information on the photophysical properties of PA matrices and doping molecules. However, before the manuscript can be recommended for publication in Nature Communications, the authors still need to address the following points:

1. The phosphorescence peak of PA6 film is assigned to be 450 nm, which is inconsistent with the experiment spectra (Figure S37a). Please explain this.
2. In the response to Q2 and Q3, the authors stated that the host-guest energy transfer play a critical role in influencing the RTP performance. However, the additional discussion might not be reasonable enough to support the conclusion. The authors must provide more convincing evidence. A comprehensive analysis based on experimental results and theoretical calculations is needed to get insight into the mechanism. Additionally, considering the fact that these doping materials are made by different preparation methods, the aggregation states of guest molecules in different matrices are suggested to be considered.
3. In the part of "Discussion on RTP Mechanism", the authors stated that "the crystalline solids of 1-4 show very weak RTP", while, in the part of "Theoretical Calculation and Discussion on Photophysical Properties", the RTP properties of 1 is characterized as "excellent". Please explain this.
4. The authors stated that the SOC matrix elements between excited states were predicted based on the single crystal structure. The H positions are very difficult to be determined by X-rays. Therefore, the optimization of H positions is required.
5. Normalized intensity is dimensionless, so "Normalized intensity / a.u." is inappropriate.

6. In the revised supporting information, “Fig. 38” and “Fig. 39” should be “Figure S38.” and “Figure S39.”. In the normalized delayed spectra of 1 @ PS in Fig.39, some values are below zero.

7. Inappropriate words lead to confusion: “delayed phosphorescence spectra”.

8. The writing of the manuscript should be improved, it was written in an extremely Chinglish style, which can indeed lead to confusion and hinder the understanding of the results.

Reviewer #4 (Remarks to the Author):

April 4, 2024

Manuscript number: NCOMMS-23-63968A

MS Type: Research Article

Title: Nylons with Highly-Bright and Ultralong Organic Room-Temperature Phosphorescence.

Correspondence Author: Dr. Yu-Wu Zhong

Responses to the review comments

Reviewer #1:

The authors addressed the raised concerns. The amended manuscript is appropriate for publishing.

Reviewer #2:

The issues have been addressed properly. I recommend publication of this paper on this journal.

Reviewer #3:

I am pleased to see that the authors have provided additional information on the photophysical properties of PA matrices and doping molecules. However, before the manuscript can be recommended for publication in Nature Communications, the authors still need to address the following points:

1. The phosphorescence peak of PA6 film is assigned to be 450 nm, which is inconsistent with the experiment spectra (Figure S37a). Please explain this.

Response: thanks for pointing out this error. The phosphorescence peak of PA6 film is indeed observed at 480 nm, which has been corrected in the revised manuscript and supplementary information.

2. In the response to Q2 and Q3, the authors stated that the host-guest energy transfer play a critical role in influencing the RTP performance. However, the additional discussion might not be reasonable enough to support the conclusion. The authors must provide more convincing evidence. A comprehensive analysis based on experimental results and theoretical calculations is needed to get insight into the mechanism. Additionally, considering the fact that these doping materials are made by different preparation methods, the aggregation states of guest molecules in different matrices are suggested to be considered.

Response: the decreased lifetimes of donor emissions in the presence of energy acceptors provide a solid evidence for the host-guest energy transfer. This is also supported by the suitable alignment of energy levels of nylons and guest chromophores. In order to provide further experimental evidence, we have performed the wavelength-dependent lifetime measurements of **2@PA6**, **3@PA6**, and **4@PA6** films. The obtained results are all in support of the host-guest energy transfer in these doped polymers. Corresponding discussions have been updated in the revised

manuscript. We tried to obtain further evidence by nanosecond transient absorption spectral analysis; however, no useful signals have been recorded to date. Theoretical calculation may provide further evidence, which however would be rather complicated considering that the energy donor in our case is a polymer film. We hope to look for collaborations in the future to perform theoretical studies on this system. For the sake of safety, the phrase of “efficient or effective energy transfer” in Abstract and Discussion was revised as “potential energy transfer” at this stage.

Regarding the aggregation states of guest molecules, a brief discussion has been added in the revised manuscript in the section of “Discussion on RTP Mechanism”: **The poor RTP performance of the PA6 film containing aggregated samples of **1** (Fig. 1g and S22) could be partially attributed to the insufficient host-guest contacts and thus less efficient energy transfer between them.** The homogeneous distribution of luminophore dopant in the polymer matrix is critical for maintaining highly-efficient RTP.

3. In the part of “Discussion on RTP Mechanism”, the authors stated that “the crystalline solids of 1-4 show very weak RTP”, while, in the part of “Theoretical Calculation and Discussion on Photophysical Properties”, the RTP properties of **1** is characterized as “excellent”. Please explain this.

Response: “the excellent RTP properties of compound **1**” has been revised as “the excellent RTP properties of **1**@PA6”.

4. The authors stated that the SOC matrix elements between excited states were predicted based on the single crystal structure. The H positions are very difficult to be determined by X-rays. Therefore, the optimization of H positions is required.

Response: the calculation details and Fig. 5 have been revised following the reviewer’s request. The H positions were optimized before the calculation of the SOC matrix elements. Corresponding discussions with some minor changes on the values of SOC matrix elements have been updated in the revised manuscript.

5. Normalized intensity is dimensionless, so “Normalized intensity / a.u.” is inappropriate.

Response: “a.u.” has been deleted in corresponding figures in the revised manuscript and supplementary materials.

6. In the revised supporting information, “Fig. 38” and “Fig. 39” should be “Figure S38.” and “Figure S39.”. In the normalized delayed spectra of **1** @ PS in Fig.39, some values are below zero.

Response: Thanks for pointing out this mistake. Corrections have been made accordingly. The phosphorescence of **1**@PS is rather weak. The below-zero values of **1**@PS in the original Figure S39 (now Fig. S40) are caused by the instrument’s background noise. This point has been indicated in the caption.

7. Inappropriate words lead to confusion: “delayed phosphorescence spectra”.

Response: “delayed phosphorescence spectra” has been corrected as “phosphorescence spectra” or “delayed emission spectra”.

8. The writing of the manuscript should be improved, it was written in an extremely Chinglish style, which can indeed lead to confusion and hinder the understanding of the results.

Response: we have carefully read the manuscript again. The English writing has been polished.

Reviewer #4:

REVIEWERS' COMMENTS

Reviewer #3 (Remarks to the Author):

The authors have addressed our concerns, and we recommend its publication now.

Reviewer #4 (Remarks to the Author):
